# TianQuan-S2S: A Subseasonal-to-Seasonal Global Weather Model via Incorporate Climatology State

**Guowen Li**[1,4], **Xintong Liu**[1], **Yang Liu**[2†], **Mengxuan Chen**[3], **Shilei Cao**[1], **Xuehe Wang**[1],
**Juepeng Zheng**[1,4†], **Jinxiao Zhang**[3], **Haoyuan Liang**[1], **Lixian Zhang**[4], **Jiuke Wang**[1],
**Meng Jin**[5], **Hong Cheng**[2], **Haohuan Fu**[3,4]

[1]Sun Yat-Sen University, [2]The Chinese University of Hong Kong, [3]Tsinghua University,
[4]National Supercomputing Center in Shenzhen, [5]Huawei Technologies Co., Ltd

## Abstract

Accurate Subseasonal-to-Seasonal (S2S) forecasting is vital for decision-making in agriculture, energy production, and emergency management. However, it remains a challenging and underexplored problem due to the chaotic nature of the weather system. Recent data-driven studies have shown promising results, but their performance is limited by the inadequate incorporation of climate states and a model tendency to degrade, progressively losing fine-scale details and yielding over-smoothed forecasts. To overcome these limitations, we propose TianQuan-S2S, a global S2S forecasting model that integrates initial weather states with climatological means via incorporating climatology into patch embedding and enhancing variability capture through an uncertainty-augmented Transformer. Extensive experiments on the Earth Reanalysis 5 (ERA5) reanalysis dataset demonstrate that our model yields a significant improvement in both deterministic and ensemble forecasting over the climatology mean, traditional numerical methods, and data-driven models. Ablation studies empirically show the effectiveness of our model designs. Remarkably, our model outperforms skillful numerical ECMWF-S2S and advanced data-driven Fuxi-S2S in key meteorological variables. The code implementation can be found in https://github.com/zhangminglang42/TianQuan.

## 1 Introduction

Subseasonal-to-Seasonal forecasting (beyond 15 days) is crucial for various applications, including agriculture, energy production, and emergency management, where it plays a vital role in planning and decision-making processes (Pegion et al., 2019). Accurate subseasonal forecasts reduce agricultural losses, balance energy supply, and improve disaster preparedness. Over the past 40 years, Numerical Weather Prediction (NWP) models have significantly advanced weather forecasting skills for the short to medium term (up to 15 days), contributing notably to Sustainable Development Goals (SDGs) (Bauer et al., 2015). Nevertheless, since the parameterization within NWP models introduces errors due to function approximations (Beljaars et al., 2018), the accumulated errors are significant on the S2S time scale. In addition, they require substantial computational resources and time (Saha et al., 2014). Even with supercomputers operating on hundreds of nodes, simulating a single variable can take hours (Bauer et al., 2020). Recently, data-driven models (Bi et al., 2023; Lam et al., 2023; Chen et al., 2023b; Price et al., 2024) present a promising future due to their comparable accuracy in medium-range forecasting and much faster inference speeds. Once trained, these models can predict future weather conditions in seconds (Pathak et al., 2022).

However, data-driven subseasonal forecasting remains challenging and less explored (Chen et al., 2024; Liu et al., 2025a). The extended time horizon renders the initial weather conditions and key variables inadequate for making accurate predictions due to the chaotic nature of the weather system (Mariotti et al., 2018). Deterministic machine learning models are typically trained on data

---

[†]Corresponding authors

from recent historical data, which provides limited support for subseasonal forecasting (He et al., 2019). As the forecast horizon extends, the initial state becomes less informative, resulting in blurred and unrealistic predictions. This limitation hampers the effectiveness of machine learning models on subseasonal time scales, despite their importance for climate-related applications (Lam et al., 2023).

Specifically, two critical issues have not been well explored in S2S forecasting: **(1) Insufficient climate modeling.** Climatology, the persistent, slow-varying climate modes that influence atmospheric conditions on the S2S time scale. While both it and initial conditions are crucial for S2S forecasting, existing approaches have predominantly focused on the former, neglecting its importance. **(2) Model collapse.** The use of discrete numerical grids in modeling inherently smooths out small-scale weather features due to spatial and temporal averaging at grid points. With extended lead times, the forecasting system progressively deteriorates, failing to preserve reliable structures and ultimately producing

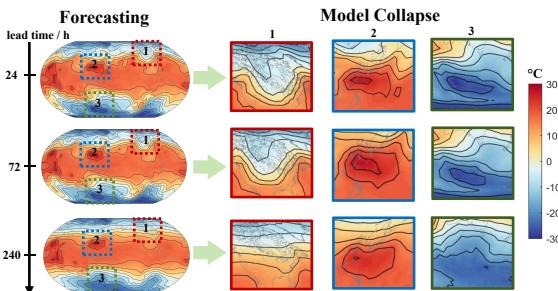

Figure 1: **Model collapse in Subseasonal-to-Seasonal (S2S) Forecasts.** As the lead time increases, the forecast results for the same target time gradually exhibit the loss of data contours across the three regions, which can be considered a form of model collapse.

unrealistic predictions. As shown in Figure 1, with the increase in lead time, the prediction of the same target days becomes smoother and finally loses predictability.

To address these issues, we propose **TianQuan-S2S**, where *TianQuan* (meaning *Weather Hub* in Chinese) reflects its role as a central system for subseasonal prediction. The framework consists of two key components to improve single-step forecasts. First, a patch embedding layer integrates both the initial weather states and climatological mean features to compensate for the limited predictive information, enhancing forecast skill beyond 15 days. Second, inspired by the traditional perturbed forecasting, which incorporates noise to enhance model performance, we extend the Vision Transformer with uncertainty blocks that inject Gaussian noise into feature representations, preventing over-smoothing and better capturing realistic variability at longer lead times. Extensive experiments are conducted on the high-quality Earth Reanalysis 5 (ERA5) dataset, showing that the proposed methods outperform skillful numerical S2S systems and data-driven models. In summary, our main contributions are as follows.

- We highlight the importance of incorporating climate information in data-driven S2S forecasting and the model collapse issue in long lead time.
- We proposed **TianQuan-S2S**, consisting of a patch embedding that incorporates climatological information and an uncertainty-augmented Transformer that captures weather variability, enhancing S2S forecasting and mitigating over-smoothing.
- Extensive experiments on the ERA5 dataset show that TianQuan-S2S outperforms traditional and data-driven methods, including ECMWF-S2S and Fuxi-S2S, in both deterministic and ensemble forecasting. Ablation studies show that the proposed two simple designs, climatology and noise incorporation, significantly enhance the model performance.

## 2 PRELIMINARIES

**Perturbed forecasting** In weather forecasting, perturbed forecasting is used to generate ensemble predictions by adding small perturbations to the initial atmospheric state (Zhou et al., 2022). Because the atmosphere is a chaotic system, even minor uncertainties in the initial conditions can lead to large differences in forecast outcomes (Nakashita & Enomoto, 2025). Perturbed forecasting generates multiple forecasts from slightly modified initial states to capture possible atmospheric evolutions. In addition to perturbing initial conditions, ensemble systems also introduce perturbations within the model itself. For example, ECMWF and NCEP apply stochastic parameterization schemes such as the Stochastically Perturbed Parametrizations (SPP), which randomly modify model tendencies like convection or turbulence (Tsiringakis et al., 2024). Moreover, data-driven methods such as SwinVRNN use learned distribution perturbations to generate diverse ensemble members, improving forecast reliability by explicitly modeling uncertainty (Hu et al., 2023).

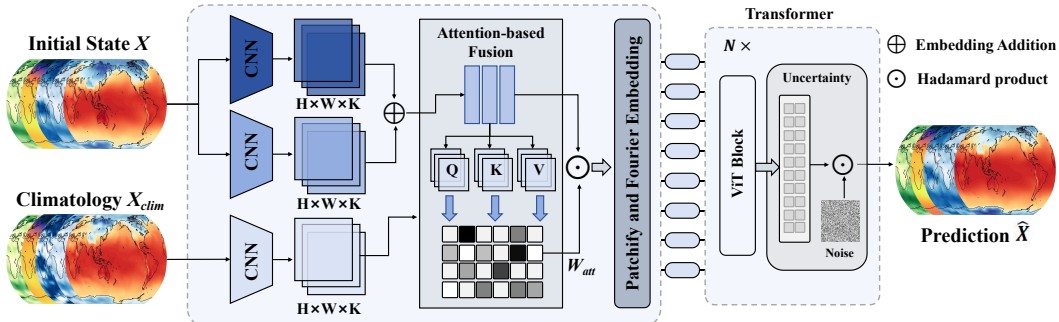

Figure 2: The diagram of TianQuan-S2S framework. The input variables include initial state $\boldsymbol{X}$ and climatology $\boldsymbol{X}_{clim}$. After attention-based fusion, the features are enhanced and fused, then patchified and fed into the uncertainty-augmented model, where Gaussian noise is injected at each layer. Finally, predictions $\hat{\boldsymbol{X}}$ for days 15 to 45 are generated.

**Climatology** Climatology is the scientific study of long-term weather patterns and atmospheric processes, typically based on decades of observational and model data. Unlike short-term forecasting, climatology emphasizes statistical averages, variability, and extremes to understand the baseline state of the climate system (White et al., 2023). It plays a central role in assessing long-term climate change, as shifts in temperature, precipitation, and circulation patterns are evaluated against climatological norms. As a reference framework, climatology provides essential context for detecting anomalies, attributing trends to natural variability or anthropogenic forcing, and guiding adaptation and mitigation strategies in response to global climate change (Forster et al., 2024). In this work, the climatology is computed based on the 38-year daily climatology from 1979 to 2016 (366 days).

**Problem Definition** We study the prediction of $K$ weather parameters at the latitude-longitude grid $\boldsymbol{G} \in \mathbb{R}^{H \times W \times 2}$, where $H$ and $W$ are the height and width of the grid that depend on the resolution of latitude and longitude, and $\boldsymbol{G}_{h,w,:} = (\lambda_h, \phi_w) \in \Omega = [-90°, 90°] \times [-180°, 180°]$. Let the state of global weather at time $t$ be represented by a 3-dimensional tensor $\boldsymbol{X}_t \in \mathbb{R}^{H \times W \times K}$, where $K = V_A \times C + V_S$, with $V_A$ denoting the number of upper-air variables, $V_S$ the number of surface variables, and $C$ the number of pressure levels. Similarly, the climatology is represented by $\boldsymbol{X}_{clim} \in \mathbb{R}^{H \times W \times K}$. Thus, our goal is to train a neural network to forecast weather variables with lead times between 15 and 45 days, as shown in the following:

$$\hat{\boldsymbol{X}}_{t_{15}:t_{45}} = f_\Theta(\boldsymbol{G}, \boldsymbol{X}_{clim}, \boldsymbol{X}_{t_{-4}:t_0}), \tag{1}$$

where $\Theta$ denotes the parameters of neural networks. $\hat{\boldsymbol{X}}_{t_{15}:t_{45}}$ is the predicted value from day 15 to day 45. In this work, we employ the 5-day historical data $\boldsymbol{X}_{t_{-4}:t_0}$ as the input, which we empirically found beneficial for model performance.

## 3 PROPOSED MODEL

Figure 2 illustrates the TianQuan-S2S framework, which consists of (1) a patch embedding that fuses the input and climatology embeddings; (2) a vision transformer that is perturbed by random noises. We illustrate their details as follows.

### 3.1 PATCH EMBEDDING

**Convolutions** Previously, some researchers (Zhang & Patel, 2018; Wang et al., 2021; Kumar & Tiwari, 2023) have utilized edge priors to enhance details. Building on this work, we introduce a multi-layer convolutional layer to extract climate information $\boldsymbol{F}_{clim} \in \mathbb{R}^{H \times W \times K}$ from the climatology $\boldsymbol{X}_{clim}$. In this convolution, pixel pair differences are calculated, and prior information is explicitly encoded into the CNN, enhancing the model's representation by learning valuable trend information.

For the initial input fields, we propose a novel convolutional design that enhances weather prediction by jointly modeling spatial and channel-wise relationships. Spatial Convolution captures regional variations in initial fields, while Channel Convolution models the relationships among variables,

such as temperature, pressure, and wind. Together, these two mechanisms generate feature maps $\boldsymbol{F_s} \in \mathbb{R}^{H \times W \times K}$ and $\boldsymbol{F_c} \in \mathbb{R}^{H \times W \times K}$, defined as follows:

$$\boldsymbol{F_s} = f_{conv}([Y_{PAP}^s, Y_{PMP}^s]), \qquad \boldsymbol{F_c} = f_{conv}(Y_{PAP}^c), \tag{2}$$

where $f_{conv}(\cdot)$ represents a neural network of convolutional layer, and $[\cdot]$ denotes channel-wise concatenation. $Y_{PAP}^c$, $Y_{PAP}^s$, and $Y_{PMP}^s$ represent features processed by partial average pooling across the channel or spatial dimension and global max pooling across the spatial dimension, respectively. Then we compute through element-wise addition to obtain the enhanced climate data as follows:

$$\boldsymbol{F_X} = \boldsymbol{F_s} + \boldsymbol{F_c} \in \mathbb{R}^{H \times W \times K}. \tag{3}$$

Note that the fusion feature information $\boldsymbol{F_X}$ can guide attention across channels, producing clearer and more refined results.

**Attention-based fusion** Research has shown that feature fusion can enhance these climate information features, aiding long-term predictions (Chen et al., 2024; Li et al., 2024). Inspired by the above, we proposed an attention-based fusion module to guide the fusion process of the features $\boldsymbol{F_{clim}}$ and $\boldsymbol{F_X}$. This module fuses two input features by first computing spatial weights that determine their relative importance. We first flatten spatial dimensions with $N = H \times W$ and fuse each variable feature with $\boldsymbol{A}^{(v)} = \mathrm{reshape}(\boldsymbol{F}_X^{(v)} + \boldsymbol{F}_{clim}^{(v)}) \in \mathbb{R}^{C \times N}$, where $C$ denotes the number of pressure levels (13 for upper-air variables and 1 for surface variables). Define:

$$\boldsymbol{Q}^{(v)} = \boldsymbol{A}^{(v)} \boldsymbol{W}_Q, \quad \boldsymbol{K}^{(v)} = \boldsymbol{A}^{(v)} \boldsymbol{W}_K, \quad \boldsymbol{V}^{(v)} = \boldsymbol{A}^{(v)} \boldsymbol{W}_V, \qquad \boldsymbol{W}_{Q/K/V} \in \mathbb{R}^{N \times N}. \tag{4}$$

Compute the spatial weights as $\boldsymbol{W}_{att} = \mathrm{unreshape}(\sigma\left(\frac{\boldsymbol{Q}^{(v)} \boldsymbol{K}^{(v)}}{\sqrt{N}}\right) \boldsymbol{V}^{(v)}) \in [0, 1]^{H \times W \times C}$. Finally, a convolutional layer maps the fused representation to the output $\boldsymbol{F} \in \mathbb{R}^{H \times W \times K}$:

$$\boldsymbol{F} = f_{conv}(\boldsymbol{F_{clim}} \cdot W_{att} + \boldsymbol{F_X} \cdot (\boldsymbol{1} - W_{att}) + \boldsymbol{F_{clim}} + \boldsymbol{F_X}), \tag{5}$$

where $\boldsymbol{1}$ denotes a matrix with all elements equal to 1. In summary, the attention-based fusion adaptively integrates complementary features, ensuring more effective fusion for long-term prediction.

**Patchify and Fourier Embedding** We divide the upper-air data of shape $C \times V_A \times H \times W$ into $L$ patches $\{\boldsymbol{F}^{(l)}\}_{l=1}^L$, where $\boldsymbol{F}^{(l)} \in \mathbb{R}^{V_A \times P \times P}$ and $L = \frac{H}{P} \times \frac{W}{P}$ denotes the number of patches. These patches are then flattened and stacked into a tensor $\boldsymbol{F}_A \in \mathbb{R}^{C \times L \times V_A \times P \times P}$, which is embedded into latent space to generate an upper embedding $\boldsymbol{E}_A \in \mathbb{R}^{C \times L \times D}$. Similarly, surface variables, influenced by terrain and the land-sea mask, are represented with $C = 1$ as $\boldsymbol{E}_S \in \mathbb{R}^{L \times D}$.

To retain spatial (latitude and longitude) and temporal information, these variables are mapped into $D$-dimensional tensors using Fourier encoding, which embeds periodic positional signals to preserve variations across space and time, implemented as follows:

$$\mathrm{FourEnc}(x) = [\sin(\frac{2\pi x}{\lambda_1}), \cos(\frac{2\pi x}{\lambda_1}), \dots, \sin(\frac{2\pi x}{\lambda_i}), \cos(\frac{2\pi x}{\lambda_i}), \dots, \sin(\frac{2\pi x}{\lambda_{D/2}}), \cos(\frac{2\pi x}{\lambda_{D/2}})] \tag{6}$$

where $\lambda_i = \lambda_{\min} \cdot \left(\frac{\lambda_{\max}}{\lambda_{\min}}\right)^{\frac{i-1}{(D/2)-1}}$ represents the wavelength of the Fourier basis function, ensuring that the encoding captures various frequency components of the input variable. For time, since climate follows a yearly cycle, the wavelength range for encoding is set with $\lambda_{\min} = 1$ and $\lambda_{\max} = 365$. For position, the wavelength range is set with $\lambda_{\min} = 0.1$ and $\lambda_{\max} = 360$, representing the spatial scale of longitude. After being processed by a linear layer, the temporal and spatial information ($\boldsymbol{W}_{pos}$, $\boldsymbol{W}_{time} \in \mathbb{R}^{(C+1) \times D}$) are integrated into initial embedding $\boldsymbol{E} \in \mathbb{R}^{(C+1) \times L \times D}$ as follows:

$$\boldsymbol{E} = [\boldsymbol{E}_S, \boldsymbol{E}_A] + \boldsymbol{W}_{pos}^* + \boldsymbol{W}_{time}^* \tag{7}$$

where $[\cdot]$ denotes C channel-wise concatenation and * denotes that broadcasting is applied during the computation. Subsequently, this embedding $\boldsymbol{E}$ is fed into the Transformer for further processing.

## 3.2 TRANSFORMER

**Transformer with noise generation** The Vision Transformer (ViT) architecture has proven effective for various prediction tasks, including climate modeling (Nguyen et al., 2023). By processing

input data with stacked attention blocks, the transformer captures both local and global dependencies. In the context of weather forecasting, each block learns hierarchical representations.

$$\hat{\boldsymbol{E}}^{(l,n)} = h_n(\boldsymbol{E}^{(l,n)}) = \text{softmax}(\frac{\boldsymbol{Q}^{(l,n)}\boldsymbol{K}^{(l,n)}}{\sqrt{D}})\boldsymbol{V}^{(l,n)} \tag{8}$$

where $\boldsymbol{Q}^{(l,n)} = \boldsymbol{E}^{(l,n)}\boldsymbol{W}_n^Q$, $\boldsymbol{K}^{(l,n)} = \boldsymbol{E}^{(l,n)}\boldsymbol{W}_n^K$, $\boldsymbol{V}^{(l,n)} = \boldsymbol{E}^{(l,n)}\boldsymbol{W}_n^B$, and $\boldsymbol{W}_m^{Q/K/V} \in \mathbb{R}^{D \times D}$ are learned projection matrices, and $n$ represents the $n$-th layer of transformer. By stacking these transformed representations, we obtain the reconstructed embedding $\hat{\boldsymbol{E}}^{(n)} = \{\hat{\boldsymbol{E}}^{(l,n)}\}_{l=1}^L$.

In general, adding stochastic perturbations helps prevent the forecast from collapsing onto a single deterministic trajectory and better captures the growth of uncertainty at extended lead times. In particular, instead of perturbing only the initial conditions, which lose impact as lead time increases (Buizza et al., 2005), we inject learnable Gaussian perturbations at each Transformer layer during the forward pass, stabilizing forecasts and mitigating over-smoothing at extended ranges.

Specifically, an uncertainty block introduces Gaussian noise at each transformer block to produce the next prediction $\boldsymbol{E}^{(n+1)}$, as described by the following formula:

$$\boldsymbol{E}^{(n+1)} = \boldsymbol{E}^{(n)} + h_n\left(\boldsymbol{E}^{(n)}\right) + g_n\left(\boldsymbol{E}^{(n)}\right) \cdot \mathcal{N}(\boldsymbol{0}, \boldsymbol{I}) \tag{9}$$

Here, $n \in \{1, \ldots, N\}$ denotes network layer index, and $h_n(.)$ represents parameters at layer $n$-th of transformer. In addition, $g_n(.)$ is a learnable parameter function introduced by the uncertainty block.

**Unpatchify** To convert the prediction embedding $\hat{\boldsymbol{E}} \in \mathbb{R}^{(C+1) \times L \times D}$ back into the prediction variables, we separate the embedding for surfce embedding $\hat{\boldsymbol{E}}_S \in \mathbb{R}^{1 \times L \times D}$ and upper-air embedding $\hat{\boldsymbol{E}}_A \in \mathbb{R}^{C \times L \times D}$. These embeddings are decoded into $P \times P$ patches via a linear layer to reconstruct the gridded data. By combining and stacking, we obtain the final output predictions, $\hat{\boldsymbol{X}}_S$ and $\hat{\boldsymbol{X}}_A$, for the target time range from day 15 to day 45.

### 3.3 LEARNING OBJECTIVES

TianQuan-S2S is trained to forecast the predictions $\hat{\boldsymbol{X}}_{t_{15}:t_{45}}$ at lead time from day 15 to day 45. The objective function used is the latitude-weighted mean squared error (Rasp et al., 2020). The loss is calculated between the prediction $\hat{\boldsymbol{X}}_{t_{15}:t_{45}}$ and the ground truth $\boldsymbol{X}_{t_{15}:t_{45}}$ as follows:

$$\mathcal{L} = \frac{1}{V \times H \times W} \sum_{v=1}^V \sum_{i=1}^H \sum_{j=1}^W L(i) \left(\hat{\boldsymbol{X}}_{t_{15}:t_{45}}^{v,i,j} - \boldsymbol{X}_{t_{15}:t_{45}}^{v,i,j}\right)^2 \tag{10}$$

in which $L(i) = \frac{\cos(\text{lat}(i))}{\frac{1}{H}\sum_{i'=1}^H \cos(\text{lat}(i'))}$ is the latitude weighting factor, and $\text{lat}(i)$ represents the latitude of the $i^{th}$ row in the grid. The coefficient size varies because grid cells near the equator cover more area than those near the poles, thus receiving higher weights.

## 4 EXPERIMENTS

**Datasets** We conducted experiments using the ERA5 dataset (Hersbach et al., 2020), covering 40 years (1979-2018) with global hourly data across multiple pressure levels and the surface. To improve the accuracy of long-term predictions, we preprocess the hourly ERA5 data by averaging it daily at six-hour intervals. Bilinear interpolation is used to downsample the data to 5.625° (32 × 64 grid points) and 1.40625° (128 × 256 grid points), while selecting the 13 most critical pressure levels. For more details on data processing, refer to Appendix F.4. We obtained the dataset variables as shown in Table 10 in Appendix F. We divided the daily averaged ERA5 dataset into training (1979-2015), validation (2016), and testing (2017-2018) sets based on years.

**Metrics** Following existing works (Bi et al., 2023; Rasp et al., 2024), we primarily quantify the performance improvement of the model using latitude-weighted RMSE and Anomaly Correlation Coefficient (ACC), and then assess its long-term forecasting capability through Continuous Ranked Probability Score (CRPS), Spread Mean Error (SME), and Relative Quantile Error (RQE). All of these metrics can be found in the Appendix G.

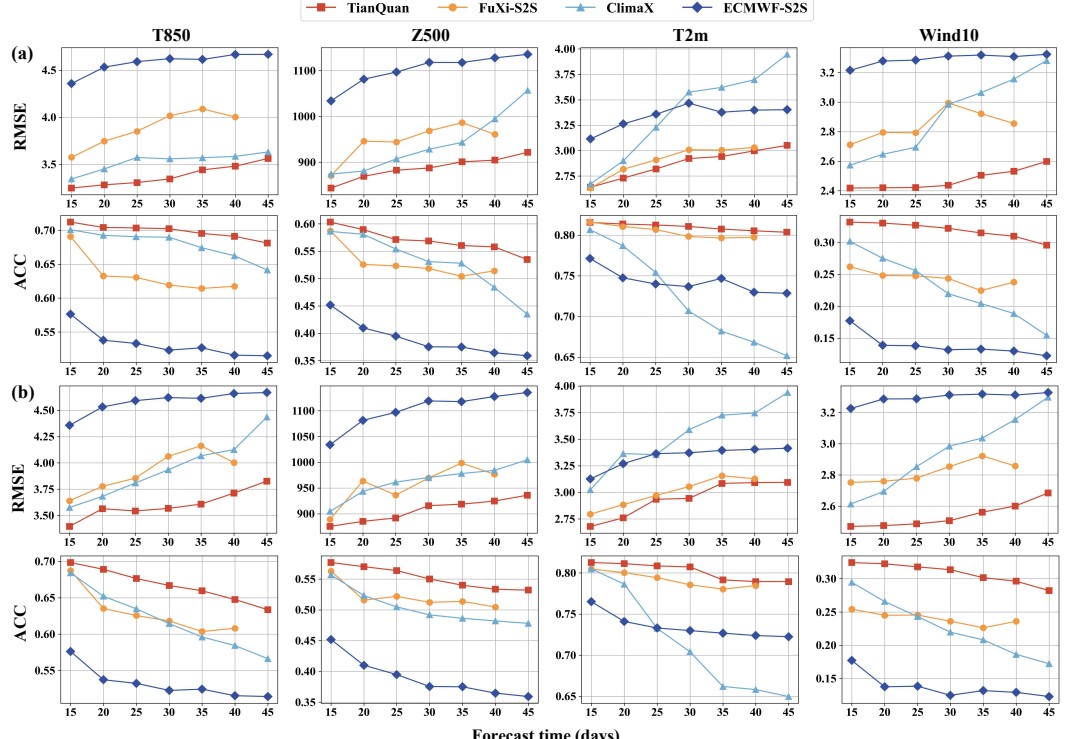

Figure 3: **TianQuan-S2S outperforms FuXi-S2S, ClimaX and ECMWF-S2S in deterministic forecasts on 1.40625°(a) and 5.625°(b) daily ERA5 datasets.** The comparison involves four variables in terms of latitude-weighted RMSE (lower is better) and ACC (higher is better). The time range for the metric calculations is from 2017 to 2018.

**Baselines** We compare our proposed model with the following baseline models:

- **ECMWF-S2S** (Vitart et al., 2017) is a WWRP/THOR PEX-WCRP joint research project established to improve forecast skill and understanding on the sub-seasonal to seasonal time scale.

- **ClimaX** (Nguyen et al., 2023) is a deep learning foundational model for weather and climate science, capable of training on ERA5 datasets with varying variables and spatio-temporal coverage. The model itself makes deterministic forecasts, and ensemble forecasts can also be obtained by adding disturbances to the initial field, as mentioned above.

- **FuXi-S2S** (Chen et al., 2024) is a state-of-the-art global subseasonal ensemble forecasting model that leverages machine learning to improve prediction accuracy, providing forecasts up to 42 days, with 40 days used here for comparison. In the comparison of deterministic forecasts, we use the FuXi-S2S (00 member) without perturbations as a contrast.

- **Climatology** utilizes the long-term average of historical data for each time step as the reference prediction, providing a simple yet effective benchmark for comparison. In this work, we refer to the Climatology Baseline standard comparison process in WeatherBench2 (Rasp et al., 2024).

We run our approach once to generate a single time value for deterministic forecasting. In addition, as an AI-based method, our model can perform large-member ensemble forecasts with small computational costs. For ClimaX, we implemented ensemble forecasting by introducing random perturbations to the initial weather states. In the case of TianQuan-S2S, we incorporated learnable noise at each block level to generate multiple forecasts. Specifically, we created 50 random perturbations, which were then added to the original unperturbed state. This approach results in a 51-member ensemble, and the ensemble mean forecast is computed by averaging the individual forecast outputs.

**Implementation Details** We used the AdamW optimizer (Kingma, 2014; Loshchilov & Hutter, 2017) with parameters $\lambda_1 = 0.9$ and $\lambda_2 = 0.99$. A weight decay of $1e^{-5}$ was applied to all parameters except the positional embeddings. The learning rate was set to $5e^{-5}$, with a linear warmup schedule

Table 1: Comparison of ensemble forecast performance across variables and models.

| Model | CRPS (↓) | | | | | | SME (↓) | | | | | | RQE (↓) | | | | | |
|---|---|---|---|---|---|---|---|---|---|---|---|---|---|---|---|---|---|---|
| | 15 | 20 | 25 | 30 | 35 | 40 | 15 | 20 | 25 | 30 | 35 | 40 | 15 | 20 | 25 | 30 | 35 | 40 |
| **T2m** ClimaX | 2.278 | 2.343 | 2.378 | 2.409 | 2.475 | 2.481 | 2.190 | 2.271 | 2.314 | 2.352 | 2.433 | 2.453 | 2.187 | 2.312 | 2.384 | 2.437 | 2.562 | 2.592 |
| FuXi-S2S | **2.238** | 2.293 | 2.328 | 2.359 | 2.426 | 2.437 | 2.140 | 2.223 | **2.264** | 2.302 | 2.383 | 2.414 | **2.136** | 2.263 | **2.338** | 2.382 | 2.513 | 2.544 |
| Ours | 2.256 | **2.276** | **2.304** | **2.297** | **2.318** | **2.341** | **2.048** | **2.141** | 2.316 | **2.235** | **2.328** | **2.338** | 2.158 | **2.244** | 2.348 | **2.331** | **2.417** | **2.469** |
| **T850** ClimaX | 2.831 | 2.827 | 2.842 | 2.833 | 2.847 | 2.866 | 2.364 | 2.354 | 2.389 | 2.379 | 2.404 | 2.414 | 2.748 | 2.815 | 2.842 | 2.883 | 2.952 | 2.967 |
| FuXi-S2S | 2.824 | 2.818 | 2.844 | 2.832 | 2.848 | 2.867 | **2.301** | 2.355 | 2.362 | 2.343 | **2.395** | 2.405 | 2.729 | **2.716** | 2.783 | **2.769** | 2.771 | 2.808 |
| Ours | **2.805** | **2.812** | **2.836** | **2.819** | **2.836** | **2.852** | 2.334 | **2.326** | **2.358** | **2.335** | 2.401 | **2.396** | **2.711** | 2.725 | **2.743** | 2.784 | **2.753** | **2.787** |
| **Wind10** ClimaX | 1.801 | 2.004 | 1.954 | 1.907 | 1.861 | 1.831 | 1.050 | 1.063 | 1.094 | 1.175 | 1.088 | 1.158 | 1.747 | 1.781 | 1.805 | 1.814 | 1.848 | 1.869 |
| FuXi-S2S | **1.798** | 1.902 | 1.963 | 1.856 | 1.852 | 1.835 | 1.038 | 1.053 | **1.065** | 1.152 | 1.075 | 1.099 | 1.736 | 1.763 | **1.813** | 1.791 | **1.803** | 1.851 |
| Ours | 1.799 | **1.901** | **1.894** | **1.852** | **1.834** | **1.826** | **1.020** | **1.034** | 1.075 | **1.128** | **1.052** | **1.081** | **1.729** | **1.754** | 1.825 | **1.779** | 1.804 | **1.834** |
| **Z500** ClimaX | 650 | 653 | 655 | 655 | 658 | 663 | 655 | 659 | 673 | 682 | 679 | 688 | 653 | 684 | 703 | 736 | 777 | 779 |
| FuXi-S2S | 645 | 648 | 655 | 653 | 654 | 661 | **634** | 649 | **659** | 663 | 670 | 669 | 634 | 665 | 693 | **705** | 718 | 723 |
| Ours | **644** | **647** | **652** | **650** | **653** | **658** | 640 | **648** | 663 | **654** | **656** | **660** | **620** | **658** | **684** | 716 | **705** | **714** |

Table 2: RMSE comparison of ensemble mean results. Our method generally achieves better performance, outperforming the climatology baseline across multiple variable metrics.

| Model | 15 | 20 | 25 | 30 | 35 | 40 | 45 | Model | 15 | 20 | 25 | 30 | 35 | 40 | 45 |
|---|---|---|---|---|---|---|---|---|---|---|---|---|---|---|---|
| **T2m** FuXi-S2S | 2.450 | **2.454** | 2.484 | 2.520 | 2.542 | 2.563 | - | **Wind10** FuXi-S2S | 3.407 | 3.574 | 3.668 | 3.678 | 3.771 | 3.752 | - |
| Ours | **2.424** | 2.457 | **2.459** | **2.502** | **2.471** | **2.532** | **2.601** | Ours | 3.345 | 3.502 | 3.595 | 3.711 | 3.753 | 3.723 | 3.742 |
| Climatology | 2.610 | 2.610 | 2.610 | 2.610 | 2.610 | 2.610 | 2.610 | Climatology | **2.430** | **2.430** | **2.430** | **2.430** | **2.430** | **2.430** | **2.430** |
| **T850** FuXi-S2S | 3.323 | 3.192 | **3.141** | 3.286 | 3.386 | 3.317 | - | **Z500** FuXi-S2S | 795 | 787 | 773 | 807 | 814 | 775 | - |
| Ours | **3.255** | **3.187** | 3.193 | **3.109** | **3.321** | **3.298** | **3.402** | Ours | **791** | **779** | **766** | **798** | **805** | **766** | **796** |
| Climatology | 3.410 | 3.410 | 3.410 | 3.410 | 3.410 | 3.410 | 3.410 | Climatology | 820 | 820 | 820 | 820 | 820 | 820 | 820 |

over 5000 steps (5 epochs), followed by a cosine annealing schedule over 95000 steps (95 epochs). Our model training was implemented using PyTorch (Paszke et al., 2019). In the training process, we train multiple models with different lead time settings, where the lead time ranges from 15 to 45 days with an interval of 5 days. The output from each model, corresponding to a single step prediction, is combined to form a 45-day subseasonal forecast target. The model was trained on eight GPUs with 80GB of memory, achieving 77.6 TFLOPS of computing power.

## 4.1 MAIN RESULTS

**Compared with deterministic forecast** We compare the ACC and RMSE of TianQuan-S2S for lead times of 15-45 days with the baselines in 4 target variables: temperature in 850hPa (**T850**), geopotential at 500hPa (**Z500**), 2m temperature (**T2m**) and 10 metre wind speed (**Wind10**) in Figure 3. Based on these results, we have the following findings:

- Regardless of whether the resolution is 1.40625° or 5.625°, TianQuan-S2S consistently outperforms all baselines in all variables. Specifically, the average RMSE improves by 0.14 on T850 ($K$), 59 on Z500 ($m^2/s^2$), and 0.353 on Wind10 ($m/s$) compared to the best baseline, all representing significant improvements.

- Compared with the transformer-based direct prediction model ClimaX, FuXi-S2S (00 member) performs worse on T850, Z500, and Wind10 before 25 days due to iterative error accumulation at each step. However, beyond 25 days, FuXi-S2S stabilizes and avoids the model collapse and severe metric degradation observed in ClimaX. Unlike ClimaX, TianQuan-S2S adds attention-based climatology fusion and learnable Gaussian noise in the Transformer blocks to anchor forecasts, reduce drift, and stabilize long-lead predictions.

- We can find that wind forecasting is more challenging for all baselines. However, TianQuan-S2S generally achieves higher accuracy across all lead times. For example, day 45 Wind ACC of ClimaX is 0.172, and ACC of ECMWF-S2S is 0.112, while ACC of TianQuan-S2S is 0.297. Under such cases, TianQuan-S2S still performs the best, further verifying its effectiveness.

**Compared with ensemble forecast** To better investigate the long-term prediction performance of TianQuan-S2S, we compare it with the ensemble forecast models in Table 1, where lighter colors are our model results. We further compare the RMSE of the ensemble mean forecast with the Climatology baseline in Table 2. From the results, we can find that:

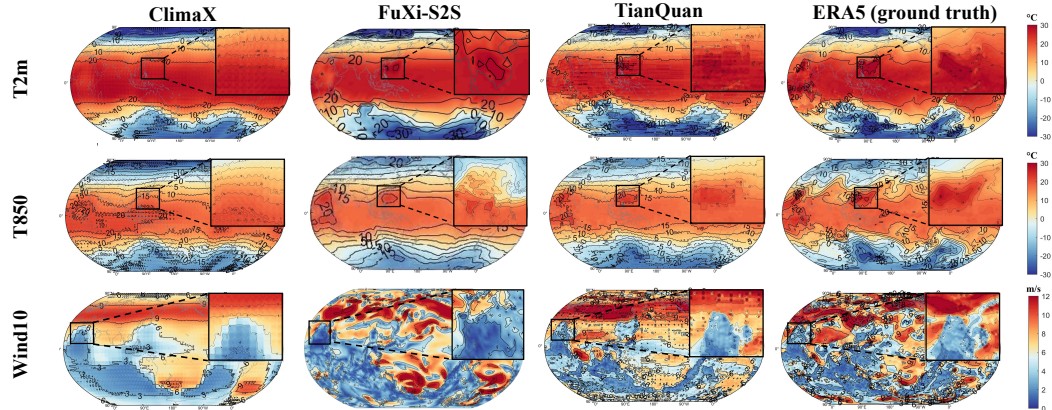

Figure 4: **Visualization of forecast results on 1.40625° daily ERA5 data.** The 30-day forecast of one upper-air variable (T850) and two surface variables (T2m and Wind10). For each case, the input time is 00:00 UTC on 15 February 2018.

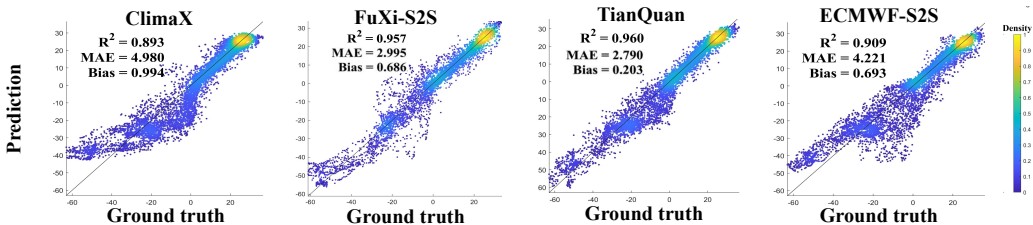

Figure 5: **Scatter chart of prediction and the ground truth.** The input time is 00:00 UTC on 15 February 2017 with the lead time of 25 days.

- Our method remarkably outperforms FuXi-S2S and ClimaX in almost all cases, demonstrating an improvement in the effectiveness of long-term predictions. In addition, the RMSE of our ensemble mean forecast remains above the Climatology baseline at T2m, T850, and Z500. However, since the average value has a greater impact on wind speed, FuXi-S2S and our approach are worse.

- We can observe that ClimaX achieves worse forecast results in ensemble forecast metrics. Also, as a direct forecast model, our method achieves significant improvements in results, particularly in the RQE of day 35 Z500 ($m^2/s^2$), which improves by 72. Such results show that models with noise generative are more stable than direct data-driven models.

- We note that for Wind10 at S2S lead times, both FuXi-S2S and our model are worse than climatology, mainly because ML-based ensemble means over-smooth this highly variable field and thus increase RMSE. In contrast, the 61-day sliding-window climatology used in WeatherBench2 (Rasp et al., 2024) yields a much smoother and more stable Wind10 baseline, especially at long leads.

**Visualization**    To provide a global view of model predictions, we visualize the day 30 prediction distribution in Figure 4. As shown in the figure, all models show a global trend of change, but the results of ClimaX have a serious loss of details. FuXi-S2S shows high value changes on both T2m and T850, but its distribution on Wind10 is significantly different from the ground truth. In contrast, our model aligns more closely with the realistic observations provided by ERA5, easing the model collapse problem. In addition, to analyze the correlation between predictions and the ground truth, we draw day 25 scatter plot in Figure 5. It demonstrates that our method is closer to the true values, and outperforms numerical model ECMWF-S2S and data-driven models, including FuXi-S2S and ClimaX, across correlation ($R^2$), mean average error (MAE), and Bias metrics. After comparing these details, we further validated our framework of incorporating climatology state by attention-based fusion. More visualizations, such as different variable predictions, can be found in the Appendix G.

## 4.2    ABLATION STUDY

**Effectiveness of the Model Design**    To validate the effect of each model design on the overall model performance, we compare four experimental models based on whether they use attention-based fusion as well as noise injection, as summarized in Table 3. The main observations are:

Table 3: Ablation studies where Clim. is short for climatology.

| | Model | RMSE (↓) | | | | | | | ACC (↑) | | | | | | |
|---|---|---|---|---|---|---|---|---|---|---|---|---|---|---|---|
| | | 15 | 20 | 25 | 30 | 35 | 40 | 45 | 15 | 20 | 25 | 30 | 35 | 40 | 45 |
| Z500 | w/o noise and Clim. | 888 | 891 | 909 | 921 | 936 | 991 | 1048 | 0.574 | 0.568 | 0.552 | 0.535 | 0.524 | 0.487 | 0.441 |
| | w/o noise | 859 | 875 | 896 | 904 | 914 | 918 | 940 | 0.591 | 0.584 | 0.563 | 0.557 | 0.542 | 0.537 | 0.521 |
| | w/o Clim. | 857 | 879 | 903 | 914 | 923 | 953 | 972 | 0.594 | 0.575 | 0.559 | 0.543 | 0.534 | 0.513 | 0.503 |
| | **Default** | **843** | **869** | **882** | **887** | **901** | **904** | **921** | **0.603** | **0.589** | **0.571** | **0.569** | **0.561** | **0.558** | **0.535** |
| T2m | w/o noise and Clim. | 3.072 | 3.145 | 3.271 | 3.591 | 3.613 | 3.716 | 3.802 | 0.790 | 0.779 | 0.750 | 0.702 | 0.684 | 0.657 | 0.648 |
| | w/o noise | 2.926 | 2.934 | 3.025 | 3.124 | 3.174 | 3.214 | 3.221 | 0.810 | 0.807 | 0.798 | 0.782 | 0.776 | 0.764 | 0.756 |
| | w/o Clim. | 3.010 | 3.072 | 3.184 | 3.224 | 3.352 | 3.471 | 3.567 | 0.804 | 0.794 | 0.775 | 0.754 | 0.735 | 0.718 | 0.708 |
| | **Default** | **2.641** | **2.734** | **2.828** | **2.922** | **2.942** | **2.998** | **3.052** | **0.816** | **0.814** | **0.812** | **0.810** | **0.807** | **0.805** | **0.796** |

Table 4: Comparison of Fusion and Perturbation Methods in Ensemble Mean RMSE. Our proposed method (Default) achieves the best performance compared to other fusion strategies, perturbation methods, and varying the number of noise injection layers.

| Methods | 15 | 20 | 25 | 30 | 35 | 40 | 45 | Methods | 15 | 20 | 25 | 30 | 35 | 40 | 45 |
|---|---|---|---|---|---|---|---|---|---|---|---|---|---|---|---|
| Concat | 2.525 | 2.556 | 2.565 | 2.583 | 2.627 | 2.655 | 2.707 | Concat | 830 | 814 | 808 | 837 | 845 | 819 | 832 |
| Gate | 2.476 | 2.528 | 2.530 | 2.541 | 2.563 | 2.589 | 2.670 | Gate | 811 | 802 | 797 | 818 | 828 | 802 | 818 |
| IC Perturb | 2.446 | 2.506 | 2.527 | 2.538 | 2.556 | 2.624 | 2.674 | IC Perturb | 814 | 812 | 809 | 821 | 823 | 808 | 818 |
| FLN | 2.497 | 2.556 | 2.560 | 2.583 | 2.601 | 2.616 | 2.617 | FLN | 825 | 814 | 818 | 827 | 825 | 816 | 823 |
| Layer 1 Only | 2.599 | 2.626 | 2.653 | 2.714 | 2.777 | 2.802 | 2.841 | Layer 1 Only | 817 | 811 | 806 | 818 | 823 | 807 | 821 |
| Layer 1-2 | 2.530 | 2.562 | 2.606 | 2.623 | 2.693 | 2.699 | 2.712 | Layer 1-2 | 810 | 809 | 798 | 816 | 818 | 792 | 811 |
| Layer 1-4 | 2.535 | 2.548 | 2.593 | 2.603 | 2.593 | 2.652 | 2.729 | Layer 1-4 | 814 | 808 | 802 | 810 | 816 | 785 | 817 |
| Layer 1-6 | 2.511 | 2.578 | 2.602 | 2.567 | 2.623 | 2.657 | 2.707 | Layer 1-6 | 807 | 797 | 772 | 811 | 812 | 779 | 815 |
| Layer 1-7 | 2.456 | 2.496 | 2.518 | 2.532 | 2.554 | 2.604 | 2.624 | Layer 1-7 | 799 | 782 | 776 | 804 | 813 | 778 | 808 |
| **Default** | **2.424** | **2.457** | **2.459** | **2.502** | **2.471** | **2.532** | **2.601** | **Default** | **791** | **779** | **766** | **798** | **805** | **766** | **796** |

(Left block labeled **T2m**, right block labeled **Z500**.)

- Comparing the models with and without the Climatology, it is evident that incorporating climate information significantly improves the metrics for forecasts beyond 25 days. For example, Z500 RMSE of the model on w/o noise and Clim. increased by 160, and its T2m RMSE increased by 0.73 from day 15 to day 45, while Z500 RMSE of the model on w/o noise only increased by 81, and its T2m RMSE increased by 0.295. This shows that climate information can serve as auxiliary information to make the model perform better in long-term predictions.

- Employing noise generation in transformer blocks improves model performance, and further gains are achieved when combined with attention-based fusion. The results in the table show that the model on w/o Clim. lowers the average RMSE of Z500/T2m by 26/0.19. In contrast, the reductions of the model on Default increase to 54/0.57. Such results not only suggest the effectiveness of our model designs but also validate the effectiveness of utilizing the transformer block with noise generation to extract the spatial periodic signal from climatology.

**Analysis of Fusion and Perturbation Methods**   We investigate our method's superiority in three areas: (1) Fusion strategies, comparing Concatenation (i.e., Concat, merging climatology on channels) and a Learnable Gate (i.e., adaptive climatology selection per pixel); (2) Perturbation methods, comparing Initial-Condition Perturbation (i.e., IC Perturb, perturbing only initial conditions) and Fixed-Layers Noise (i.e., FLN, removing learnable functions); (3) Impact of noise injection layers, where "Layers 1-2" refers to injecting noise into the 1st and 2nd blocks. Results in Table 4 show that:

- Comparing the two variants of fusion strategies and ensemble forecast perturbations on T2m and Z500 at all lead times, our method (Default) achieves the best performance, confirming that attention is a more effective and interpretable way to exploit climatological priors, and indicating that the proposed learnable noise injection is a superior alternative to Initial-Condition Perturbations and better emulates the climate evolution process than Fixed-Layer Noise.

- Investigating the impact of uncertainty blocks, we further conduct an ablation study on per-layer perturbation. As the number of perturbed layers increases, the overall performance consistently improves: the ensemble-mean RMSE of T2m and Z500 is reduced by up to 0.223 and 28.85, respectively, demonstrating that noise injection enhances forecast accuracy.

**Impact of Noise Scale**   To further understand the impact of the standard deviation of Gaussian noise, we compared different noise standard deviations $\sigma$ and obtained the results shown in Figure 6. From the results, we can find that when $\sigma$ is around 1, the perturbation model generates better results, indicating that this noise scale strikes an optimal balance between regularization and model accuracy.

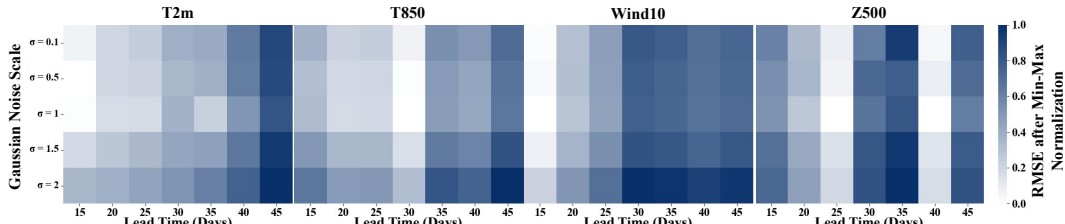

Figure 6: RMSE comparison between different Gaussian noise scales at all lead times. A lighter color indicates better results: overall, the effect is better when $\sigma = 1$.

## 5 RELATIVE WORK

**Data-driven weather forecasts** Weather forecasting mainly involves gridded prediction tasks, akin to image-to-image translation. Recent advancements (Zheng et al., 2025) have been made in short-to medium-term weather and climate forecasting. Notable examples include FourCastNet (Pathak et al., 2022), which was the first to attempt multi-variable forecasting in the short term with results comparable to NWP models. Pangu-Weather (Bi et al., 2023) and GraphCast (Lam et al., 2022) surpass traditional numerical methods in achieving multi-variable forecasting in the medium term. Based on the framework of transformer, FuXi (Chen et al., 2023b) and FengWu (Chen et al., 2023a) developed a model method adapted to the Earth system. NeuralGCM (Kochkov et al., 2024) is designed based on the GCM architecture to capture spatial dependencies in weather prediction. Exploring higher resolution forecasting, FengWu-GHR (Han et al., 2024) aims to improve prediction accuracy. Aurora (Bodnar et al., 2025) and Prithvi-WxC (Schmude et al., 2024) both focus on Weather forecasting based on foundation models. Aardvark Weather (Allen et al., 2025) and FuXi Weather (Sun et al., 2025) both employ end-to-end deep learning models, directly processing raw observational data and assimilating satellite information for weather forecasting. While these advancements extend forecasting capabilities, the absence of effective climate information leads to a significant decline in performance. This highlights the need for new approaches specifically designed for subseasonal forecasting.

**Subseasonal-to-Seasonal forecasts** Subseasonal forecasting, which provides predictions 2 to 6 weeks ahead, fills the crucial gap between short-term weather forecasts (typically up to 15 days) and long-term climate predictions that extend to seasonal and longer timescales. Although machine learning models have achieved significant progress in medium-range and climate prediction, their effectiveness in subseasonal forecasting has been less pronounced (Wang et al., 2024; He et al., 2021). However, at sub-seasonal to seasonal time scales, the predictive results of machine learning methods often lack the details of forecasting (Vitart et al., 2022; Nathaniel et al., 2024). This limitation leads to iterative errors in long-term forecasts, ultimately rendering the results unusable (Lam et al., 2023). Recent methods focus on building more effective subseasonal forecasting models to enhance predictive skill (Chen et al., 2024; Liu et al., 2025a;b). To strengthen subseasonal forecasting capability, we proposed a patch embedding that incorporates climatological information and an uncertainty-augmented Transformer that captures weather variability. In particular, although both Fuxi-S2S and TianQuan-S2S add perturbations in the model, Variational Autoencoder-based FuXi-S2S adds perturbations only within the latent space, while TianQuan-S2S injects learnable Gaussian noise directly into the feature representations at every layer of Transformer blocks, continuously reinforces variability, and more effectively mitigates model collapse in long-lead forecasts. In addition, Fuxi-S2S does not explicitly incorporate climate information.

## 6 CONCLUSION AND FUTURE WORK

In this work, we present TianQuan-S2S, a novel framework that addresses the challenges in subseasonal-to-seasonal (S2S) forecasting, particularly the limitations of model collapse and the insufficient use of climate information. Our extensive experiments on the ERA5 dataset demonstrate that TianQuan-S2S outperforms both traditional numerical S2S systems and data-driven models, providing significant improvements in both deterministic and ensemble forecasting. Ablation studies have further substantiated the effectiveness of model designs, and additional empirical analysis illustrates the superior performance across all lead times. In the future, we plan to optimize our model framework to enable higher-resolution forecasts and incorporate additional information such as land, ocean, and sea ice data to improve the learning capabilities of the model.

## 7 ACKNOWLEDGMENTS

This work was supported in part by the National Natural Science Foundation of China under Grant T2125006; in part by GuangDong Basic and Applied Basic Research Foundation under Grant 2025A1515510016; in part by Shenzhen Science and Technology Program under Grant KCXFZ20240903093759004 and Grant KJZD20230923115106012, in part by the Open Research Fund of Pengcheng Laboratory under No.2025KF1B0040. Yang Liu is supported in part by the Postdoctoral Fellowship Scheme of The Chinese University of Hong Kong.

## 8 REPRODUCIBILITY STATEMENT

We are committed to ensuring the reproducibility of our research. Source code and a README.md file with detailed instructions for data preparation and script execution are available at `https://github.com/zhangminglang42/TianQuan` and also provided in the Appendix. The appendix offers comprehensive details to support our claims. Appendix C describes the details and hyperparameters of our proposed TianQuan-S2S. In Appendix D, we further explored the differences in the prediction methods of each model and conducted extensive experiments on the prediction methods of our own model. Furthermore, Appendix E and F list in detail the sources and processing methods of the experimental data, and the calculation formulas of the indicators are also included.

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

# TianQuan-S2S: A Subseasonal-to-Seasonal Global Weather Model via Incorporate Climatology State (APPENDIX)

TABLE OF CONTENTS IN APPENDIX

APPENDIX

## A  THE USE OF LARGE LANGUAGE MODELS (LLMS)

In this study, Large Language Models (LLMs) were used as an assistive tool to enhance the overall quality and clarity of the text. The primary research, analysis, and intellectual contributions remain solely the work of the authors. The key applications of LLMs in this study are as follows:

- **Text Refinement:** LLMs were employed to enhance the grammatical accuracy, structure, and consistency of the manuscript. This involved refining sentence flow and ensuring consistent phrasing and tone across the entire paper.
- **Coherence and Structure:** LLMs were used to organize and strengthen the logical flow of ideas. Suggestions were incorporated to improve transitions between sections and paragraphs, making the narrative more cohesive.
- **Idea Articulation:** At various stages, LLMs acted as a sounding board, helping to articulate complex ideas more clearly and explore alternative expressions for the concepts already proposed by the authors.

All recommendations made by the LLM were critically reviewed, revised, and approved by the authors to ensure accuracy and alignment with the research intent. The final responsibility for the content lies solely with the authors.

## B  NOTATIONS

Table 5 summarizes the notations appearing in this paper.

Table 5: Summary of key notations.

| Symbol | Description |
| --- | --- |
| $\boldsymbol{G}$ | Latitude-longitude grid with dimensions $H \times W \times 2$ |
| $\boldsymbol{X}_t, \boldsymbol{X}_{clim}$ | the state of global weather at time $t$ and climatology $\mathbb{R}^{H \times W \times K}$ |
| $K$ | Total number of weather variables, calculated as $V_A \times C + V_S$ |
| $C$ | Number of pressure levels |
| $V_A, V_S$ | Number of upper-air variables, surface variables |
| $\hat{\boldsymbol{X}}_{t_{15}:t_{45}}$ | Predicted weather variables from day 15 to day 45 |
| $\boldsymbol{F_{clim}}$ | Convolutional feature map extracted from climatology $\boldsymbol{X}_{clim}$ |
| $\boldsymbol{F_s}, \boldsymbol{F_c}$ | Spatial and channel feature maps from convolution operations |
| $\boldsymbol{F_X}$ | Enhanced climate feature map after fusion of $\boldsymbol{F_s}$ and $\boldsymbol{F_c}$ |
| $\boldsymbol{A}^{(v)}$ | Reshaped feature tensor after flattening spatial dimensions |
| $\boldsymbol{Q}^{(v)}, \boldsymbol{K}^{(v)}, \boldsymbol{V}^{(v)}$ | Query, key, and value matrices for attention mechanism |
| $\boldsymbol{W}_{Q/K/V}$ | Learnable projection matrices for query, key, and value |
| $\boldsymbol{W}_{att}$ | Attention weights for spatial fusion |
| $\boldsymbol{E}, \boldsymbol{E}_A, \boldsymbol{E}_S$ | Combined and upper-air and surface embedding tensors after patchify |
| $\hat{\boldsymbol{E}}^{(l,n)}$ | Transformer output after applying attention mechanism |
| $g_n(.)$ | Learnable noise function in the uncertainty block |
| $\mathcal{L}, \mathrm{lat}(i)$ | Latitude-weighted mean squared error loss function, and latitude of the $i$-th grid row |

## C   MODEL DETAILS

This section presents the improved details of TianQuan-S2S.

### C.1   IMPROVED DETAILS

After comparing our proposed TianQuan-S2S with FuXi-S2S and ClimaX in the experiments, we identified the following details as critical for subseasonal forecasts performance:

- **Multi-scale Temporal Modeling**: Capturing features at different time scales is crucial for subseasonal forecasts. Using multi-scale convolutions or multi-layer recurrent neural networks can effectively handle both short-term and long-term dependencies.
- **Incorporation of Attention Mechanisms**: Self-attention mechanisms can dynamically assign importance to different time steps, allowing the model to focus more on the time steps that have a greater impact on future predictions, thereby improving accuracy.
- **Climate change trend information**: For time series with significant seasonality and long-term climate trends, enhancing and fusing climate features separately can significantly improve prediction performance.

### C.2   HYPERPARAMETERS

The hyperparameters of the model are shown in Table 6.

Table 6: Hyperparameters and their meanings for the model.

| Hyperparam | Meaning | Value |
|:---:|:---|:---:|
| $\lvert \mathcal{V} \rvert$ | Number of default variables | 67 |
| $D$ | Embedding dimension | 384 |
| Depth | Number of UD-ViT blocks | 8 |
| heads | Number of attention heads | 12 |
| Drop path | For stochastic depth | 0.1 |
| Dropout | Dropout rate | 0.12 |

## D   INFERENCE DETAILS

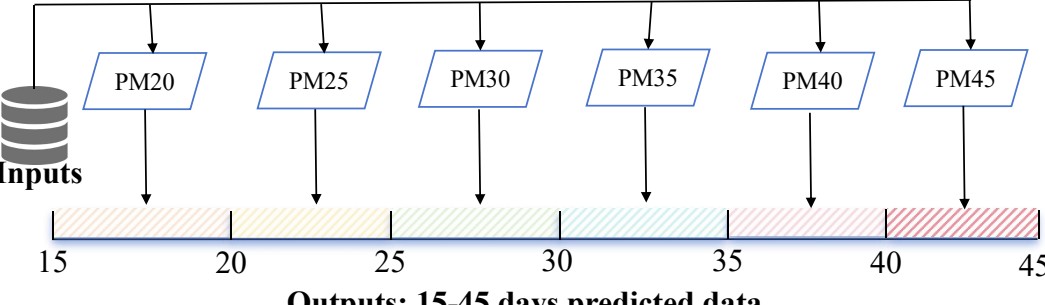

**Inputs**

**Outputs: 15-45 days predicted data**

Figure 7: **Multi-model forecasting strategy.** PM$\mathcal{K}$ denotes forecast models with $\mathcal{K}$-day lead times. With at least 5 days of input data, predictions can be made for the following 15-45 days.

Our approach differs from hierarchical temporal aggregation methods used in Pangu-Weather (Bi et al., 2023). We divide the 15–45 day prediction horizon into 5-day segments and train a separate model $PM_{\mathcal{K}}$ for each. Formally, this design can be written as:

$$X_{t-4+K, ..., t+K} = PM_K\left(X_{t-4, ..., t}\right), \ t \in R \tag{11}$$

where $\mathcal{K} = \{20, 25, ..., 45\}$ represents the lead time of the prediction model. Each model generates its prediction solely from the initial fields spanning time steps $t - 4$ to $t$, without relying on outputs from earlier models, thereby preventing the accumulation of errors across models. Meanwhile, we proposed the Content Fusion Module to integrate auxiliary signals (e.g., t+n climatology) and utilize the robust backbone (UD-ViT) for improved long-range modeling. It can improve the accuracy of each segment.

### D.1 FORECAST STRATEGY RESEARCH

At present, the methods of weather forecasting are mainly divided into autoregressive forecasting and multi-model single-step combination forecasting.

To help distinguish the predictions of currently common weather forecast models, we compare the models of Pangu-Weather (Bi et al., 2023), FuXi (Chen et al., 2023b), and ClimaX (Nguyen et al., 2023).

Table 7: Comparison of representative forecasting models.

| Model | Pretrain | Single-Model AR | Horizon Gran. | Horizon |
|---|---|---|---|---|
| Pangu-Weather | × | × | 1h, 3h, 6h, 24h | Up to 7d |
| FuXi | ✓ (ERA5) | ✓ (Cascade) | Each model 5d | Up to 15d |
| ClimaX | ✓ (CMIP6) | ✓ (ClimaX-iter) | 1, 3, 5, 7, 10, 14, 30d | Up to 30d |
| TianQuan-S2S | × | × | 15, 20, ..., 45d (step=5) | 15–45d |

We have also implemented a single autoregressive model fine-tuned by scheduled sampling based on our framework. The RMSE comparison results are shown in the following table:

Table 8: Comparison of multi-model and single autoregressive model.

| Strategy | Vars | 5 | 10 | 15 | 20 | 25 | 30 | 35 | 40 | 45 |
|---|---|---|---|---|---|---|---|---|---|---|
| multi-model | T2m | 1.83 | 2.82 | 2.64 | 2.73 | 2.82 | 2.922 | 2.942 | 2.998 | 3.052 |
| | T850 | 1.73 | 2.95 | 3.249 | 3.283 | 3.307 | 3.346 | 3.442 | 3.481 | 3.563 |
| | Wind10 | 2.03 | 2.37 | 2.4185 | 2.4203 | 2.4219 | 2.4368 | 2.5043 | 2.5321 | 2.5984 |
| | Z500 | 3.40 | 7.80 | 843.6 | 869.2 | 882.6 | 887.4 | 900.8 | 904.5 | 921.6 |
| single auto-regressive model | T2m | 1.52 | 2.67 | 2.79 | 2.89 | 2.95 | 3.25 | 3.41 | 3.61 | 3.80 |
| | T850 | 1.82 | 3.12 | 3.32 | 3.412 | 3.5142 | 3.6142 | 3.7814 | 3.941 | 4.023 |
| | Wind10 | 1.94 | 2.26 | 2.465 | 2.501 | 2.583 | 2.623 | 2.771 | 2.817 | 3.062 |
| | Z500 | 3.10 | 7.74 | 875.1 | 888.9 | 899.3 | 935.4 | 948.2 | 967.1 | 985.6 |

As shown in the table 8, the autoregressive strategy outperforms the multi-model strategy in most metrics for lead times under 10 days. However, for longer-range forecasts (beyond 10 days), the multi-model strategy consistently achieves better performance across key metrics, highlighting its superiority in the subseasonal-to-seasonal forecasting domain.

## E DISCUSSION

### E.1 CONSISTENCY ANALYSIS OF MULTI-MODEL OUTPUTS

To address the concern regarding temporal discontinuities between adjacent lead times, our model primarily analyzes the continuity of the output at the two key points:

- **Residual connections for consistent representations**: Residual connections in the Transformer preserve information across layers, helping the model maintain coherent representations from short to long leads and reducing abrupt shifts in the forecasts.

- **Smooth transitions and stability**: Beyond climatology fusion, we add lead-time and absolute-time embeddings (days since 1 Jan 1970) so the model is explicitly aware of the target time, which improves temporal continuity and stability between adjacent lead-time forecasts.

Additionally, we further assess temporal continuity by computing the first-order differences between adjacent model outputs and analyzing mean/std/max in the Table 9.

Table 9: Mean, standard deviation, and maximum of first-order differences for different variables and lead-time ranges.

| Metrics | Vars | 15–20 | 20–25 | 25–30 | 30–35 | 35–40 | 40–45 |
|---------|------|-------|-------|-------|-------|-------|-------|
| **mean** | T2m | 0.132 | 0.152 | 0.187 | 0.210 | 0.286 | 0.316 |
| | T850 | 0.199 | 0.206 | 0.227 | 0.239 | 0.302 | 0.329 |
| | Wind10 | 0.0138 | 0.0189 | 0.0196 | 0.0255 | 0.0353 | 0.0531 |
| | Z500 | 13.01 | 13.96 | 16.47 | 16.60 | 18.15 | 27.90 |
| **std** | T2m | 1.973 | 1.986 | 2.432 | 2.596 | 2.603 | 2.716 |
| | T850 | 2.558 | 2.657 | 2.921 | 3.228 | 3.257 | 3.527 |
| | Wind10 | 0.971 | 1.128 | 1.180 | 1.377 | 1.414 | 1.497 |
| | Z500 | 176.2 | 173.7 | 203.5 | 182.3 | 204.5 | 226.2 |
| **max** | T2m | 11.88 | 13.01 | 15.36 | 16.45 | 17.57 | 18.79 |
| | T850 | 17.43 | 22.09 | 27.16 | 28.22 | 28.75 | 28.82 |
| | Wind10 | 6.852 | 7.331 | 8.595 | 9.913 | 12.33 | 12.36 |
| | Z500 | 1031 | 1078 | 1252 | 1361 | 1322 | 1482 |

Note: **K-(K+5)** denotes the continuity metric between the K-day and (K+5)-day model predictions.

**Overall**, the mean, standard deviation, and maximum of the first-order differences increase with lead time across all variables, consistent with the gradual degradation of forecast skill at longer ranges. **At the same time**, the growth of these statistics across adjacent lead-time windows is smooth rather than abrupt, indicating that the fields evolve in a temporally continuous manner instead of exhibiting erratic, frame-to-frame jumps.

### E.2 CHALLENGE IN WIND FORECASTING

The challenge in accurately forecasting near-surface variables, such as Wind10, primarily stems from the extreme complexity of the underlying surface conditions and strong thermodynamic processes. Factors like heterogeneous terrain, land-sea contrast, and surface heating induce intense turbulence and high variability, making near-surface wind inherently less predictable than large-scale atmospheric variables.

We acknowledge that current data-driven models, including FuXi-S2S and TianQuan-S2S, are currently limited by the absence of high-resolution surface data and explicit solar radiation inputs, which are essential for capturing these fine-scale processes. This limitation explains why simple climatology, which smooths out subseasonal variability, can sometimes appear more stable in evaluation metrics for Wind10.

We fully agree on the importance of these inputs. The current model is evaluated primarily on lower-resolution data and does not yet incorporate other crucial Earth system components (e.g., land surface, ocean, and sea ice data), which significantly influence subseasonal climate variability.

We plan to extend the model to support higher-resolution inputs and integrate multi-source Earth observation data (including detailed surface and radiation information) to enhance its physical consistency and forecasting skill.

Table 10: Dataset Variable List for Different Types and Levels.

| Type | Variable name | Abbrev. | Levels |
|------|---------------|---------|--------|
| Static | Land-sea mask | LSM | - |
| Static | Orography | - | - |
| Surface | 2 metre temperature | T2m | - |
| Surface | 10m Wind speed | $Wind_{10}$ | - |
| Upper | Geopotential | Z | 50, 100, 150 |
| Upper | Wind speed | Wind | 200, 250, 300 |
| Upper | Temperature | T | 400, 500, 600 |
| Upper | Specific humidity | Q | 700, 850 |
| Upper | Relative humidity | R | 925, 1000 |

## F EMPLOYED DATA

In this paper, two pre-generated datasets, ERA5 and ECMWF-S2S, are primarily utilized. The following sections will introduce their sources and respective roles.

### F.1 ERA5

We used the ERA5 (Rasp et al., 2020) dataset to complete the model training process. ERA5 was created as a standard benchmark dataset and evaluation framework for comparing data-driven weather forecasting models. WeatherBench regridded the original ERA5 from 0.25° to three lower resolutions: 5.625° and 1.40625°. The corresponding data can be downloaded from https://cds.climate.copernicus.eu/cdsapp#!/dataset/reanalysis-era5-complete?tab=overview.

### F.2 ECMWF-S2S

The Sub-seasonal to Seasonal Prediction Project (S2S) (Vitart et al., 2017), initiated in November 2013 by the World Weather Research Programme (WWRP) and the World Climate Research Programme (WCRP), aims to enhance forecast skill and deepen our understanding of the dynamics and climate drivers on the sub-seasonal to seasonal timescale (ranging from two weeks to a season). This project seeks to bridge the gap between medium-range and seasonal forecasting by leveraging the combined expertise of the weather and climate research communities, thereby addressing critical issues relevant to the Global Framework for Climate Services (GFCS). We can download its data from https://apps.ecmwf.int/datasets/data/s2s/

### F.3 FUXI-S2S

The forecast results of FuXi-S2S were downloaded from Hugging Face to facilitate a direct comparison with our model. The data, available at https://huggingface.co/FuXi-S2S, were used to evaluate the performance of our approach in forecasting subseasonal-to-seasonal (S2S) predictions. This comparison allowed us to assess the forecast accuracy and skill across various variables, such as temperature and precipitation, in order to better understand the relative strengths and weaknesses of our model.

### F.4 DATA PROCESSING

Since our task is performed on ERA5 daily average data, we simply introduce this data processing here. We preprocess the hourly ERA5 data by sampling every six hours (00, 06, 12, 18 UTC+00) and then computing daily averages over these four time points. For example, the 00-day average represents the sum of the average values from 23 to 00 on the previous day. The calculation is

expressed as follows:

$$Var_{t_0} = \frac{1}{T} \sum_{t=t_0-T+1}^{t_0} Var_t, T = 24 \tag{12}$$

where $Var$ represents other variables except wind speed. For wind speed, the daily mean is calculated as:

$$\text{Wind}_{t_0} = \frac{1}{T} \sum_{t=t_0-T+1}^{t_0} \sqrt{u_t^2 + v_t^2}, \quad T = 24 \tag{13}$$

where $u_t$ and $v_t$ are the zonal and meridional wind components at each sampled time.

## G QUANTITATIVE EVALUATION

In this section, we detail the evaluation metrics applied in our experiments. The predictions and ground truth are formatted as $N \times H \times W$, where $N$ indicates the quantity of predictions or test samples, and $H \times W$ defines the spatial resolution. To address the varying sizes of grid cells, $L(i)$ is introduced as a latitude-based weighting factor.

**Root mean square error (RMSE)** Root mean square error (RMSE) measures the average magnitude of the prediction errors in a dataset. It quantifies how closely predicted values match the actual values, with lower RMSE indicating more accurate predictions.

$$\text{RMSE} = \frac{1}{N} \sum_{k=1}^{N} \sqrt{\frac{1}{H \times W} \sum_{i=1}^{H} \sum_{j=1}^{W} L(i)(\tilde{y}_{k,i,j} - y_{k,i,j})^2} \tag{14}$$

**Anomaly correlation coefficient (ACC)** Anomaly Correlation Coefficient (ACC) assesses the accuracy of a forecast by measuring the correlation between predicted and observed anomalies. It is commonly used in climate and weather forecasting to evaluate how well the model captures deviations from the climatological average. Higher ACC values indicate better predictive skill.

$$\text{ACC} = \frac{\sum_{k,i,j} L(i)\tilde{Y}'_{k,i,j} Y'_{k,i,j}}{\sqrt{\sum_{k,i,j} L(i)\tilde{Y}'^2_{k,i,j} \sum_{k,i,j} L(i)Y'^2_{k,i,j}}} \tag{15}$$

$$\tilde{Y}'_{k,i,j} = \tilde{y}'_{k,i,j} - C_{k,i,j}, \quad y'_{k,i,j} = y'_{k,i,j} - C_{k,i,j} \tag{16}$$

where Climatology C represents the average of the ground truth data across the entire test set over time.

**R-squared (R²), Mean Absolute Error (MAE) and Bias** R-squared (R²), Mean Absolute Error (MAE) and Bias are commonly used to assess the performance of predictive models. The metrics are shown in Figure 5.

$$R^2 = 1 - \frac{\sum_{i=1}^{N} (y_i - \hat{y}_i)^2}{\sum_{i=1}^{N} (y_i - \bar{y})^2} \tag{17}$$

$$MAE = \frac{1}{N} \sum_{i=1}^{N} |y_i - \hat{y}_i| \tag{18}$$

$$Bias = \frac{1}{N} \sum_{i=1}^{N} (\hat{y}_i - y_i) \tag{19}$$

**Continuous Ranked Probability Score (CRPS)** CRPS measures the difference between the forecast CDF and the step function at the observed value, thus evaluating both the sharpness and reliability of a probabilistic forecast. A lower CRPS indicates a more accurate forecast. **Calculation Formula:**

$$\text{CRPS}(F, y) = \int_{-\infty}^{+\infty} \left( F(x) - \mathbf{1}\{x \geq y\} \right)^2 dx, \tag{20}$$

where $F(x)$ is the forecast cumulative distribution function (CDF) and $\mathbf{1}\{x \geq y\}$ is the indicator function for the observed value $y$.

**Spread Mean Error (SME)**   SME quantifies the difference between the ensemble spread and the absolute error of the ensemble mean. This metric is useful to determine whether the ensemble is under-dispersive (negative SME) or over-dispersive (positive SME), and thus helps in assessing the reliability of the ensemble prediction system.**Calculation Formula:**

$$\text{SME} = \frac{1}{N} \sum_{i=1}^{N} \left( \sigma_i - |y_i - \mu_i| \right), \tag{21}$$

where for each case $i$, $\sigma_i$ is the ensemble spread (e.g., the standard deviation), $\mu_i$ is the ensemble mean, and $y_i$ is the observed value.

**Relative Quantile Error (RQE)**   RQE measures the relative error between the forecast quantiles and the observation across different probability levels. It helps to assess how well the forecast distribution captures the observed outcome. A lower RQE indicates that the forecast quantiles are in closer agreement with the observations, implying a more skillful probabilistic forecast. **Calculation Formula:**
Assuming $K$ quantile levels $\alpha_k$, RQE is defined as:

$$\text{RQE} = \frac{1}{K} \sum_{k=1}^{K} w_k \left| \frac{q_{\alpha_k} - y}{y} \right|, \tag{22}$$

where $q_{\alpha_k}$ denotes the forecast quantile at probability level $\alpha_k$, $y$ is the observed value, and $w_k$ are the weights (with $\sum_{k=1}^{K} w_k = 1$).

## H    ADDITIONAL RESULTS

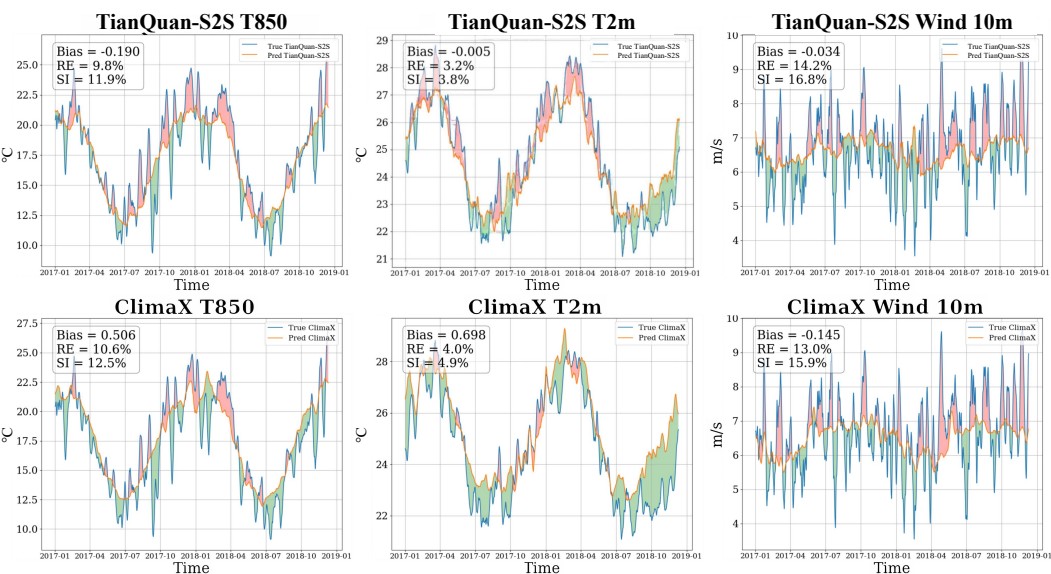

Figure 8: **Fixed-point time series diagrams for TianQuan-S2S and ClimaX across three variables (T850, T2m, and Wind**$_{10}$**).** The time series covers the period from 2017 to 2018, with latitude and longitude fixed at 23° and 113°.

Figure 8 primarily illustrates the differences between pointwise time series predictions and actual values on 5.625° data. The results show that both TianQuan-S2S and ClimaX, as machine learning methods, can effectively capture the temperature trends associated with seasonal changes, demonstrating the advantages of machine learning in climate forecasting. Although there is still some deviation in more complex variables like 10m wind speed, both methods achieve relatively stable forecasting. Notably, TianQuan-S2S outperforms ClimaX in long-term sequence metrics such as RE and SI on the variable of temperature, but ClimaX is better on the variable of wind.

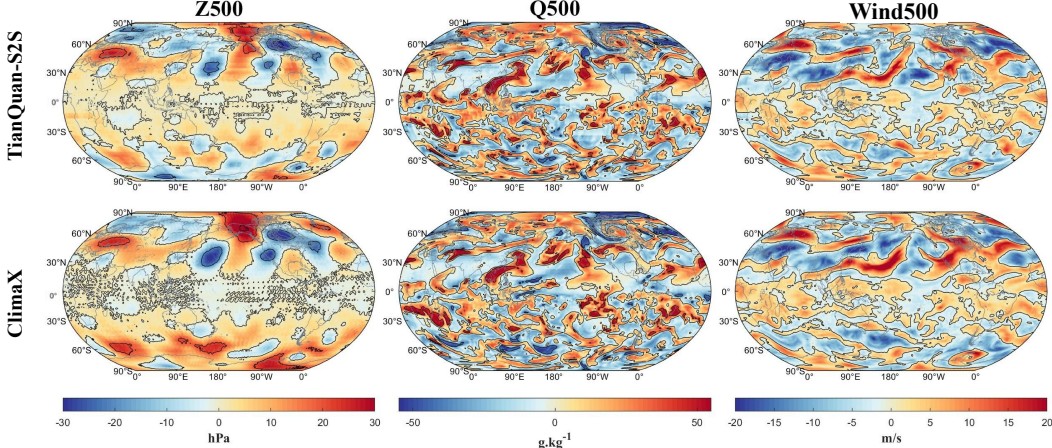

Figure 9: **Comparison of TianQuan-S2S and ClimaX in predicting deviations for three atmospheric variables: Z500, Q500, and Wind500**. The top row shows deviations from TianQuan-S2S, and the bottom row from ClimaX.

The Figure 9 illustrates the deviation of predictions from actual values for Z500, Q500, and W500 variables using TianQuan-S2S and ClimaX. Overall, TianQuan-S2S exhibits more accurate predictions with smaller and more evenly distributed deviations across all three variables, indicating

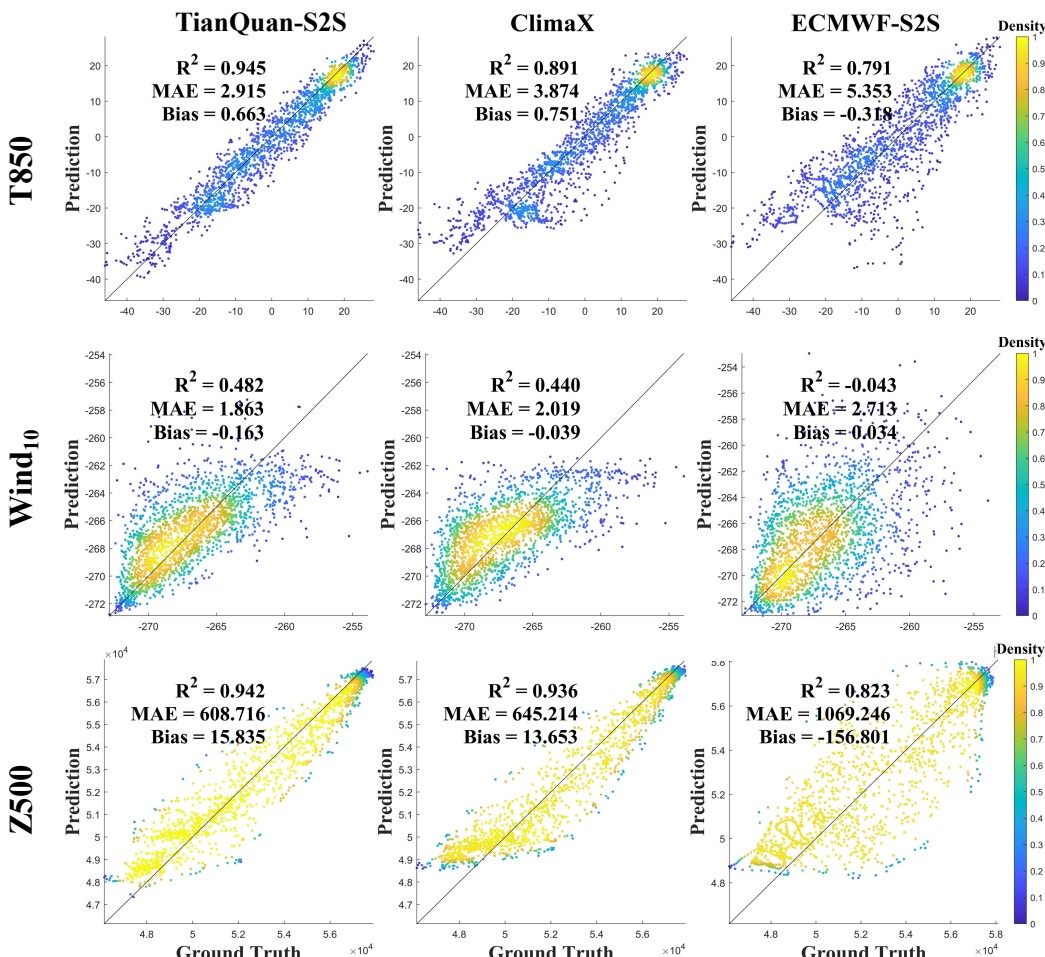

Figure 10: **The scatter plots in the figure compare the performance of TianQuan-S2S, ClimaX, and ECMWF-S2S across three variables: T850, Wind10, and Z500.** Each model's predictions are plotted against the ground truth, with density colors indicating the concentration of data points. The performance metrics, including R², MAE, and Bias, are provided for each model.

better alignment with the actual atmospheric patterns. In contrast, ClimaX shows larger deviations, particularly in areas with more complex dynamics, suggesting it struggles to capture finer details and variations. The results demonstrate TianQuan-S2S's superior capability in minimizing prediction errors across different atmospheric conditions.

The results shown in Figure 10 indicate the following: For the Wind10 and Z500 variables, all models show a reduction in predictive accuracy compared to T850, with lower R² values across the board. Despite this, TianQuan-S2S consistently outperforms the other models. It achieves the highest R² values for both Wind10 (0.482) and Z500 (0.942), along with the lowest MAE values (1.863 and 608.716, respectively), indicating superior precision. While ClimaX performs reasonably well, its accuracy is slightly diminished, as reflected in its higher MAE and marginally lower R² values. ECMWF-S2S, however, struggles significantly with both variables, displaying minimal correlation with the ground truth, especially in Wind10, where it shows almost no predictive capability. The results confirm TianQuan-S2S's robustness across varying atmospheric conditions, while ClimaX and ECMWF-S2S show limitations in handling more complex variables like Wind10 and Z500.

In addition, we have referred to the prior works and included a new comparison with traditional ML models, including XGBoost and Lasso, for forecasting T2m within the 15-45 day range.

Table 11: Comparison of TianQuan-S2S, XGBoost, and Lasso across different forecast horizons.

| Models | 15 | 20 | 25 | 30 | 35 | 40 | 45 |
|---|---|---|---|---|---|---|---|
| TianQuan-S2S | 2.64 | 2.73 | 2.82 | 2.92 | 2.94 | 2.99 | 3.05 |
| XGBoost | 2.82 | 2.99 | 3.14 | 3.29 | 3.42 | 3.52 | 3.61 |
| Lasso | 3.05 | 3.22 | 3.38 | 3.53 | 3.65 | 3.74 | 3.79 |

As shown in the Table 11, while these models perform reasonably well, our proposed TianQuan-S2S consistently achieves lower RMSE, particularly at longer lead times. This demonstrates the superiority of our architecture in S2S forecasting.

Table 12: T2m RMSE of ablation studies for North America and East Asia.

| Regional | Model | 15 | 20 | 25 | 30 | 35 | 40 | 45 |
|---|---|---|---|---|---|---|---|---|
| **North American** | w/o noise and Clim. | 3.108 | 3.226 | 3.345 | 3.632 | 3.711 | 3.843 | 3.853 |
| | w/o noise | 3.095 | 3.144 | 3.262 | 3.342 | 3.406 | 3.604 | 3.611 |
| | w/o Clim. | 2.980 | 3.027 | 3.103 | 3.203 | 3.250 | 3.332 | 3.312 |
| | Default | **2.715** | **2.764** | **2.913** | **3.007** | **3.048** | **3.113** | **3.123** |
| **East Asia** | w/o noise and Clim. | 3.037 | 3.151 | 3.275 | 3.554 | 3.577 | 3.751 | 3.797 |
| | w/o noise | 2.951 | 2.943 | 3.039 | 3.102 | 3.153 | 3.188 | 3.253 |
| | w/o Clim. | 2.966 | 3.099 | 3.160 | 3.201 | 3.332 | 3.440 | 3.545 |
| | Default | **2.591** | **2.686** | **2.834** | **2.919** | **2.886** | **2.965** | **3.011** |

Additionally, we have supplemented our analysis with ablation experiments across two different regions to further investigate the performance of our model in different places. Based on Table 12, we provide a comprehensive discussion:

- Due to more complex orography and land–sea contrast, T2m RMSE over North America is overall higher than over East Asia; moreover, in North America, the "w/o Clim." case (only noise) is better than "w/o noise" (only climatology), indicating that noise perturbations bring larger gains than climatology in more complex terrain.

- From 15 to 45 days, the advantage of the Default over all ablations persists and generally enlarges, confirming that our components are especially helpful for mitigating long-lead skill degradation while still improving short leads.

Table 13: T2m RMSE of ablation studies for North America and East Asia.

| Variables | Model | 15 | 20 | 25 | 30 | 35 | 40 | 45 |
|---|---|---|---|---|---|---|---|---|
| **T850** | w/o noise and Clim. | 3.867 | 3.923 | 3.883 | 3.990 | 4.006 | 4.062 | 4.116 |
| | w/o noise | 3.599 | 3.614 | 3.691 | 3.846 | 3.875 | 3.982 | 3.931 |
| | w/o Clim. | 3.412 | 3.495 | 3.573 | 3.666 | 3.666 | 3.698 | 3.860 |
| | Default | **3.249** | **3.283** | **3.307** | **3.346** | **3.442** | **3.481** | **3.563** |
| **Wind10** | w/o noise and Clim. | 2.844 | 2.861 | 2.894 | 2.815 | 2.919 | 2.922 | 3.110 |
| | w/o noise | 2.694 | 2.719 | 2.756 | 2.793 | 2.804 | 2.903 | 2.923 |
| | w/o Clim. | 2.658 | 2.598 | 2.558 | 2.670 | 2.748 | 2.694 | 2.795 |
| | Default | **2.418** | **2.420** | **2.422** | **2.437** | **2.504** | **2.532** | **2.598** |

From Table 13, for different variables (T850 and Wind10), the Default model (with noise + climatology) consistently achieves the lowest RMSE across all leads, confirming that our design benefits variables with diverse physical scales.

To investigate performance during challenging conditions, we analyzed the model's results during the severe 2018 North American (NAM) and 2018 East Asian (EA) heatwaves, starting from June 28, 2018, and July 9, 2018, respectively in Table 14.

Table 14: T2m RMSE of ablation studies for North America and East Asia.

| Lead Time | Model | NAM | EA |
|---|---|---|---|
| **25-day** | ClimaX | 2.948 | 3.046 |
| | FuXi-S2S | 2.634 | 2.851 |
| | TianQuan-S2S | **2.522** | **2.721** |
| **40-day** | ClimaX | 3.081 | 3.120 |
| | FuXi-S2S | 2.724 | 2.942 |
| | TianQuan-S2S | **2.685** | **2.843** |

We can find that TianQuan-S2S achieves better results compared to existing models (ClimaX, FuXi-S2S) even under these extreme cases, demonstrating its practical usefulness. While these results are encouraging, we acknowledge that improvements on extreme-weather metrics are more modest than for bulk statistics, suggesting that rare, high-impact events in complex terrain remain more challenging and warrant further targeted investigation.

Table 15: T2m RMSE Comparison of Ensemble Mean Results for Different Noise Scale.

| Regional | Model | 15 | 20 | 25 | 30 | 35 | 40 | 45 |
|---|---|---|---|---|---|---|---|---|
| **Input Perturbations** | $\sigma = 0.1$ | 2.509 | 2.554 | 2.568 | 2.569 | 2.624 | 2.630 | 2.718 |
| | $\sigma = 0.5$ | 2.477 | 2.535 | 2.539 | 2.543 | 2.569 | **2.592** | 2.679 |
| | $\sigma = 1$ | **2.446** | **2.506** | **2.527** | **2.538** | **2.556** | 2.624 | **2.674** |
| | $\sigma = 1.5$ | 2.551 | 2.580 | 2.594 | 2.614 | 2.664 | 2.692 | 2.734 |
| | $\sigma = 2$ | 2.611 | 2.637 | 2.642 | 2.652 | 2.696 | 2.710 | 2.774 |
| **Fixed Noise** | $\sigma = 0.1$ | 2.507 | **2.538** | **2.548** | **2.553** | 2.607 | **2.614** | **2.696** |
| | $\sigma = 0.5$ | 2.505 | 2.556 | 2.565 | 2.583 | 2.617 | 2.635 | 2.707 |
| | $\sigma = 1$ | **2.497** | 2.556 | 2.560 | 2.583 | **2.601** | 2.616 | 2.719 |
| | $\sigma = 1.5$ | 2.615 | 2.638 | 2.648 | 2.663 | 2.695 | 2.717 | 2.786 |
| | $\sigma = 2$ | 2.681 | 2.708 | 2.721 | 2.728 | 2.762 | 2.798 | 2.886 |

Notably, the results shown in Table 15, the performance of Input Perturbation and Fixed Noise rapidly drops when the noise scale increases to 1.5.

