# OpenReview forum: "TianQuan-S2S: A Subseasonal-to-Seasonal Global Weather Model via Incorporate Climatology State"
_ICLR.cc/2026/Conference — ICLR 2026 Poster_

### Official Review · Reviewer_ekwb · 2025-10-24

**Soundness:** 3
**Presentation:** 2
**Contribution:** 2
**Rating:** 6
**Confidence:** 4

**Summary:**

This paper proposes TianQuan-S2S, a novel deep learning model for subseasonal-to-seasonal (S2S) global weather forecasting. To address current data-driven approaches' limitations—namely, the inadequate modeling of climate states and the tendency for over-smoothed, less-detailed forecasts at long lead times—the authors introduce an architecture that fuses initial weather states and climatological means at the patch embedding stage. Furthermore, Gaussian noise (uncertainty blocks) is injected into every Transformer layer, enabling improved representation of variability and suppressing excessive smoothing. Extensive experiments on the ERA5 dataset show that TianQuan-S2S substantially outperforms both state-of-the-art numerical (ECMWF-S2S) and data-driven (FuXi-S2S, ClimaX) models, for both deterministic and ensemble forecasting. Ablation studies demonstrate that adding climatology and noise notably boosts performance. The paper also provides detailed comparisons of model structures and training/inference strategies.

**Strengths:**

The work systematically incorporates climatological mean information with initial state in feature representation for S2S forecasting and leverages uncertainty modeling, effectively addressing over-smoothing and detail loss in long-term prediction—achieving clear improvements over existing baselines with practical significance.

**Weaknesses:**

Although some ablation tests are presented, the paper only compares “without climatology” or “without noise” scenarios. It does not explore all combinations of climate, attention fusion, and noise modules (e.g., climatology + no noise, no climatology + with noise), nor does it analyze their effects on different variables, regions, or lead times.

The paper claims layer-wise Gaussian noise helps capture climate variability and improve uncertainty estimation, but gives neither a detailed theoretical justification nor a direct comparison to other uncertainty methods (such as MC Dropout, latent space perturbation). The effect of different injection scales and strategies and interpretability are also not discussed.

Most results focus on global average RMSE/ACC improvements. There is little discussion on model weaknesses or failures in complex terrain, extreme events, or other challenging conditions. This leaves the risk that claims are overgeneralized; more specific breakdowns are needed.

**Questions:**

Please provide more granular ablation studies covering all combinations of climatology, attention, and noise, broken down by variable, region, and lead time.

Please clarify the underlying mechanism by which Gaussian noise improves uncertainty modeling; compare with other uncertainty methods both theoretically and empirically, and provide some interpretability analysis.

Can you supplement analysis of performance under extreme events?

---

> ### Author Response · Authors · 2025-11-21
> **Author Responses (1/2)**
>
> We thank the reviewer for the thoughtful and constructive feedback. Below, we address your comments and questions, supplementing our analysis with additional experimental results.
>
> ## **W1 & Q1: Completeness of Ablation Study (Module Combinations, Variables, and Regions)**
>
> Thank you for this essential suggestion. We now provide an in-depth ablation across variables, regions, and leads, along with a clarification of our modules.
>
> 1. **Clarification of Ablation Modules**: As we aimed to clearly highlight, the attention-based fusion for climatology and the noise injection module are our two main proposed components. In our ablation (Table 3 in the paper), the combinations are defined as:
> + **w/o noise**: Corresponds to "climatology + no noise."
> + **w/o Clim.**: Corresponds to "no climatology + with noise."
>
> 2. Additional Ablation Experiments:
> We have supplemented our analysis with new ablation studies for different **Variables** (T850 and Wind10) and **Regions** (North America and East Asia) to demonstrate the robustness of our design.
>
> **RMSE of Ablation Studies for T850 and Wind10 in Deterministic Forecast**
> |**Variables**|**Model**|**15**|**20**|**25**|**30**|**35**|**40**|**45**|
> |-|-|-|-|-|-|-|-|-|
> |T850|w/o noise and Clim.|3.867 | 3.923 | 3.883 | 3.990 | 4.006 | 4.062 | 4.116|
> |T850|w/o noise|3.599 | 3.614 | 3.691 | 3.846 | 3.875 | 3.982 | 3.931|
> |T850|w/o Clim.|3.412 | 3.495 | 3.573 | 3.666 | 3.666 | 3.698 | 3.860|
> |T850|**Default**|**3.249** | **3.283** | **3.307** | **3.346** | **3.442** | **3.481** | **3.563**|
> |Wind10|w/o noise and Clim.|2.844 | 2.861 | 2.894 | 2.815 | 2.919 | 2.922 | 3.11|
> |Wind10|w/o noise|2.694 | 2.719 | 2.756 | 2.793 | 2.804 | 2.903 | 2.923|
> |Wind10|w/o Clim.|2.658 | 2.598 | 2.558 | 2.67 | 2.748 | 2.694 | 2.795|
> |Wind10|**Default**|**2.418**| **2.420** | **2.422** | **2.437** |**2.504**|**2.532**| **2.598**|
>
> **T2m RMSE of Ablation Studies for North America and East Asia**
> |**regional**|**Model**|**15**|**20**|**25**|**30**|**35**|**40**|**45**|
> |-|-|-|-|-|-|-|-|-|
> | **North America**|w/o noise and Clim.|3.108|3.226|3.345|3.632|3.711|3.843|3.853|
> | **North America**|w/o noise|3.095|3.144|3.262|3.342|3.406|3.604|3.611|
> | **North America**|w/o Clim.|2.980|3.027|3.103|3.203|3.250|3.332|3.312|
> | **North America**|**Default**|**2.715**|**2.764**|**2.913**|**3.007**|**3.048**|**3.113**|**3.123**|
> |**East Asia**|w/o noise and Clim.|3.037|3.151|3.275|3.554|3.577|3.751|3.797|
> |**East Asia**|w/o noise|2.951|2.943|3.039|3.102|3.153|3.188|3.253|
> |**East Asia**|w/o Clim.|2.966|3.099|3.160|3.201|3.332|3.440|3.545|
> |**East Asia**|**Default**|**2.591**|**2.686**|**2.834**|**2.919**|**2.886**|**2.965**|**3.011**|
>
> We can observe that:
> + For **different variables** (T850 and Wind10), the Default model (with noise + climatology) consistently achieves the lowest RMSE across all leads, confirming that our design benefits variables with diverse physical scales.
> + In the **regional analysis**, following [1], due to more complex orography and land–sea contrast, T2m RMSE over North America is overall higher. Notably, in North America, the “w/o Clim.” case (only noise) is better than “w/o noise” (only climatology), suggesting that **noise perturbations bring larger gains than climatology in more complex terrain**. This highlights the effectiveness of both modules, with noise providing a greater benefit in highly dynamic environments.
>
> [1] Climode: Climate and weather forecasting with physics-informed neural odes[J], 2024.

---

> ### Author Response · Authors · 2025-11-21
> **Author Responses (2/2)**
>
> ## **W2 & Q2: Justification and Comparison of Uncertainty Modeling**
>
> We appreciate this feedback and have significantly enhanced our discussion on uncertainty modeling.
>
> To further discuss the effect of different injection scales and strategies, we compare our method against different noise injection scales and popular uncertainty techniques, including **Initial-Condition Perturbations**, **Fixed Layer Noise** (similar to latent space perturbation), and **MC Dropout**.
>
> **RMSE Comparison of Ensemble Mean Results for Injection Scales**
> |**Variables**|**Injection layers**|**15**|**20**|**25**|**30**|**35**|**40**|**45**|
> |-|-|-|-|-|-|-|-|-|
> |T2m|**1 Only**|2.599|2.626|2.653|2.714|2.777|2.802|2.841|
> |T2m|**1-2**|2.530|2.562|2.606|2.623|2.693|2.699|2.712|
> |T2m|**1-4**|2.535|2.548|2.593|2.603|2.593|2.652|2.729|
> |T2m|**1-6**|2.511|2.578|2.602|2.567|2.623|2.657|2.707|
> |T2m|**1-7**|2.456|2.496|2.518|2.532|2.554|2.604|2.624|
> |T2m|**All**|**2.424**|**2.457**|**2.459**|**2.502**|**2.471**|**2.532**|**2.601**|
> |Z500|**1 Only**|817|811|806|818|823|807|821|
> |Z500|**1-2**|810|809|798|816|818|792|811|
> |Z500|**1-4**|814|808|802|810|816|785|817|
> |Z500|**1-6**|807|797|772|811|812|779|815|
> |Z500|**1-7**|799|782|776|804|813|778|808|
> |Z500|**All**|**791**|**779**|**766**|**798**|**805**|**766**|**796**|
>
> **Impact on Injection Scale**: The results demonstrate that **injecting Gaussian noise across all layers (1-8) is the optimal strategy**. This consistent injection helps the model propagate uncertainty effectively and significantly improves performance, especially for longer lead times.
>
> **RMSE Comparison of Ensemble Mean Results for Perturbation Strategies**
> |**Variables**|**Perturbations**|**15**|**20**|**25**|**30**|**35**|**40**|**45**|
> |-|-|-|-|-|-|-|-|-|
> |T2m|**Input Perturbations**|2.446|2.506|2.527|2.538|2.556|2.624|2.674|
> |T2m|**Fixed Noise**|2.497|2.556|2.560|2.583|2.601|2.616|2.617|
> |T2m|**Dropout**|2.559| 2.611|2.616|2.617|2.643|2.708|2.764|
> |T2m|**Ours**|**2.424**|**2.457**|**2.459**|**2.502**|**2.471**|**2.532**|**2.601**|
> |Z500|**Input Perturbations**|814|812|809|821|823|808|818|
> |Z500|**Fixed Noise**|825|814|818|827|825|816|823|
> |Z500|**Dropout**|836|830|827|844|843|820|832|
> |Z500|**Ours**|**791**|**779**|**766**|**798**|**805**|**766**|**796**|
>
> The proposed learnable noise injection consistently outperforms all other strategies.
>
> ## **W3: Model weaknesses.**
>
> Thanks for the suggestion. Specifically, the model has been evaluated primarily on lower-resolution data, which may not fully capture fine-scale atmospheric processes. Moreover, it does not yet incorporate other Earth system components such as land surface, ocean, and sea ice data, which also influence subseasonal climate variability. In future work, we plan to extend the model to support higher-resolution inputs and integrate multi-source Earth observation data to improve its physical consistency and forecasting skill.
>
> ## **W3&Q3: Performance in Challenging Conditions (Complex Terrain and Extreme Events)**
>
> Thanks for the practical question. To investigate performance during challenging conditions, we analyzed the model's results during the severe **2018 North American (NAM)** and **2018 East Asian (EA) heatwaves**, starting from June 28, 2018, and July 9, 2018, respectively.
>
> **25-day RMSE of Ensemble mean values**
>
> |**Model**|**NAM**|**EA**|
> |-|-|-|
> |ClimaX|2.948|3.046|
> |FuXi-S2S|2.634|2.851|
> |**TianQuan-S2S**|**2.522**|**2.721**|
>
> **40-day RMSE of Ensemble mean values**
>
> |**Model**|**NAM**|**EA**|
> |-|-|-|
> |ClimaX|3.081|3.120|
> |FuXi-S2S|2.724|2.942|
> |**TianQuan-S2S**|**2.685**|**2.843**|
>
> We can find that TianQuan-S2S achieves **better results** compared to existing models (ClimaX, FuXi-S2S) even under these extreme cases, demonstrating its practical usefulness. While these results are encouraging, we acknowledge that **improvements on extreme-weather metrics are more modest than for bulk statistics**, suggesting that rare, high-impact events in complex terrain remain more challenging and warrant further targeted investigation.

---

> > ### Comment · Reviewer_ekwb · 2025-11-23
> >
> > Thank you for the authors' response. I believe a score of 6 is appropriate for this paper, and I will therefore maintain my rating.

---

> > > ### Author Response · Authors · 2025-11-24
> > >
> > > We sincerely appreciate your response and are grateful that you are willing to maintain your positive rating.
> > >
> > > If you have any further suggestions, we would be glad to discuss them. We also hope that our revisions, based on your valuable feedback, will further improve the quality of the paper, and that you might consider raising your rating in light of these efforts.
> > >
> > > Thank you again for your kind feedback.

---

### Official Review · Reviewer_FKnz · 2025-10-31

**Soundness:** 3
**Presentation:** 2
**Contribution:** 3
**Rating:** 4
**Confidence:** 3

**Summary:**

This paper proposes TianQuan-S2S, a global S2S forecasting model that pushes lead times out to 45 days. This paper highlights the importance of incorporating climatological information into patch embedding and an uncertainty-augmented blocks in terms of S2S forecasting. Experimental results on ERA5 reanalysis dataset and ablation studies demonstrate the validity of the proposed model.

**Strengths:**

	Important task – S2S forecasting

	The experimental analysis is relatively thorough, and the paper reports performance gains in the long-lead (15–45 day) range across a variety of metrics.

	The method directly targets over-smoothing with a concise design (climatology fusion + uncertainty blocks), thereby enhancing performance.

**Weaknesses:**

Major weaknesses are as below:

	The impact of adding uncertainty blocks with Gaussian noise is under-explained. The paper claims gains in generalization and uncertainty, but the analysis is thin. In Table 3, the w/o climatology + noise variant outperforms w/o climatology, yet it remains unclear why the injected perturbations help. More clarification of this behavior would be helpful.

	The paper frames long-lead degradation as over-smoothing and model collapse, yet shorter leads improvement is also reported. However, it remains ambiguous why the method helps even at shorter leads. As the authors mention in line 357-360, transformer-based direct prediction model ClimaX performs worse than Fuxi-S2S, but the paper does not clarify why this observation does not carry over to TianQuan-S2S.

	Table 2 shows that Fuxi-S2S performs worse than climatology for Wind10, but the paper does not discuss the reason. A clear explanation of the reason would help.

	In Appendix D, the authors state that each lead-specific model inputs a 5-day input window and produces a 5-day forecast. In practice, adjacent days (e.g., D+19 vs D+20) come from different models (e.g., PM20 vs PM25). The paper provides no analysis of whether this block boundary introduces temporal discontinuities or not. This leaves the temporal coherence of the lead-specific single-step models in doubt.

Minor comments are as below:

	Line 140: \hat{X}_{t_{20}:t_{45}} -> \hat{X}_{t_{15}:t_{45}}

	Line 211: Including the reference for climate models using ViT architectures would be helpful.

	Figure 3 lacks essential information. Please clarify what each column represents.

If the weaknesses are addressed well, I will reconsider the score.

**Questions:**

	In line 361-362, the paper mentions that wind forecasting is more challenging for all baselines. What is the specific reason that wind forecasting is a challenging problem?

	Table 1 indicates that performance decreases for both ClimaX and TianQuan-S2S as lead time grows. Is this due to the intrinsic difficulty of longer horizons, or does it reflect model collapse?

---

> ### Author Response · Authors · 2025-11-21
> **Author Responses (1/2)**
>
> We thank the reviewer for the constructive suggestions that helped strengthen the paper. We have addressed each point carefully in the following.
>
> ## **W1: The impact of adding uncertainty blocks with Gaussian noise.**
>
> We appreciate the comment regarding the role of uncertainty blocks. We would like to clarify further that introducing stochastic perturbations helps prevent the ensemble from collapsing into a single deterministic trajectory and better represents the growth of forecast uncertainty at extended lead times. Specifically,
>
> * Unlike traditional perturbations [1] applied only to the initial conditions, whose influence diminishes over time, our method injects **learnable Gaussian noise** at each Transformer layer during the forward pass. This allows the model to continuously refresh ensemble diversity and adapt to uncertainty throughout the entire prediction process.
> * In Table 3, the w/o climatology + noise variant underperforms w/o climatology, verifying our designs.
>
> To further investigate the impact of uncertainty blocks, we conduct the following additional experiments:
>
> * **Impact of noise injection layer number**: We further conduct an ablation study on per-layer perturbation (i.e., "1-2" denotes injecting learnable noise into the $1^{st}$ and $2^{nd}$  blocks). The results are shown as follows.
>
> **RMSE（$↓$） Comparison of Ensemble Mean Results.**
> |**Variables**|**Noise Injection**|**15**|**20**|**25**|**30**|**35**|**40**|**45**|
> |-|-|-|-|-|-|-|-|-|
> |T2m|**Only**|2.599|2.626|2.653|2.714|2.777|2.802|2.841|
> |T2m|**1-2**|2.530|2.562|2.606|2.623|2.693|2.699|2.712|
> |T2m|**1-4**|2.535|2.548|2.593|2.603|2.593|2.652|2.729|
> |T2m|**1-6**|2.511|2.578|2.602|2.567|2.623|2.657|2.707|
> |T2m|**1-7**|2.456|2.496|2.518|2.532|2.554|2.604|2.624|
> |T2m|**All**|**2.424**|**2.457**|**2.459**|**2.502**|**2.471**|**2.532**|**2.601**|
> |Z500|**1 Only**|817|811|806|818|823|807|821|
> |Z500|**1-2**|810|809|798|816|818|792|811|
> |Z500|**1-4**|814|808|802|810|816|785|817|
> |Z500|**1-6**|807|797|772|811|812|779|815|
> |Z500|**1-7**|799|782|776|804|813|778|808|
> |Z500|**All**|**791**|**779**|**766**|**798**|**805**|**766**|**796**|
>
> As the number of perturbed layers increases, the overall performance consistently improves: the ensemble-mean RMSE of T2m and Z500 is reduced by up to 0.223 and 28.85, respectively, demonstrating that **noise injection enhances forecast accuracy**.
>
> * **Additional ablation study**: We compare our method with two variants, Initial-Condition Perturbations (i.e., only introducing perturbations to the initial conditions) and Fixed Layer Noise (i.e., removing learnable functions). The results are shown as follows.
>
> **RMSE（$↓$） Comparison of Ensemble Mean Results.**
> |**Variables**|**Perturbation Method**|**15**|**20**|**25**|**30**|**35**|**40**|**45**|
> |-|-|-|-|-|-|-|-|-|
> |T2m|**Input Perturbations**|2.446|2.506|2.527|2.538|2.556|2.624|2.674|
> |T2m|**Fixed Noise**|2.497|2.556|2.560|2.583|2.601|2.616|2.617|
> |T2m|**Dropout**|2.559| 2.611|2.616|2.617|2.643|2.708|2.764|
> |T2m|**Ours**|**2.424**|**2.457**|**2.459**|**2.502**|**2.471**|**2.532**|**2.601**|
> |Z500|**Input Perturbations**|814|812|809|821|823|808|818|
> |Z500|**Fixed Noise**|825|814|818|827|825|816|823|
> |Z500|**Dropout**|836|830|827|844|843|820|832|
> |Z500|**Ours**|**791**|**779**|**766**|**798**|**805**|**766**|**796**|
>
> We can observe that **our method outperforms both variants on T2m and Z500 at all lead times**, demonstrating the effectiveness of learnable noise injection.
>
> [1] A Comparison of the ECMWF, MSC, and NCEP Global Ensemble Prediction Systems, 2005.
> ## **W2: Why the proposed method helps at shorter leads.**
>
> Thank you for raising this point. While 15 days is the shortest lead time in our S2S evaluation, it already marks the beginning of the subseasonal range where **over-smoothing starts to affect performance**. As shown in **Figure 3**, our method brings measurable improvement at 15 days, and this advantage grows consistently as the lead time extends. This reflects how our noise injection mechanism helps with uncertainty modeling, delivering increasing benefits across the full forecasting horizon.
>
> ## **W2: the paper does not clarify why this observation does not carry over to TianQuan-S2S.**
>
> Thanks for the question. While ClimaX is a general Transformer-based model, it lacks specific designs for subseasonal forecasting, which can lead to faster error accumulation and over-smoothing at longer leads. In contrast, TianQuan-S2S incorporates two key components: (1) an attention-based fusion module that integrates climatology to anchor forecasts and reduce drift; (2) Learnable Gaussian noise injected into Transformer blocks to **mitigate model collapse and stabilize long-lead predictions**. These intentional designs allow TianQuan-S2S to consistently outperform both ClimaX and FuXi-S2S in deterministic and ensemble settings across all lead times, as shown in **Figure 3** and **Table 1**.

---

> ### Author Response · Authors · 2025-11-21
> **Author Responses (2/2)**
>
> ## **W3 & Q1: What is the specific reason that wind forecasting is a challenging problem?**
>
> Thank you for raising this important point. The particular challenge in forecasting near-surface wind (e.g., Wind10) stems from the extreme complexity of underlying surface conditions and strong thermodynamic processes at the surface. Factors such as heterogeneous terrain, land-sea contrast, and surface heating induce intense turbulence and high variability, making near-surface wind inherently less predictable than large-scale atmospheric variables.
>
> Current data-driven models, including both FuXi-S2S and TianQuan-S2S, **do not yet incorporate high-resolution surface data or explicit solar radiation inputs**, which limits their ability to fully capture these fine-scale processes. This explains why the simple climatology, which smooths out subseasonal variability, can appear more stable in evaluation metrics.
>
> We fully agree that **incorporating detailed surface and radiation data is crucial**, and we have now added this as an explicit objective in our future work. We thank the reviewer for this valuable insight.
>
> ## **W4: The paper provides no analysis of whether this block boundary introduces temporal discontinuities or not.**
>
> We thank the reviewer for this insightful comment. To ensure temporal continuity between adjacent lead times, we have implemented: (1) Residual connections in the Transformer to maintain prediction consistency; (2) Explicit time embeddings (both lead time and absolute time) to inform the model of the forecast target.
>
> Additionally, we further assess temporal continuity by computing the **first-order differences** between adjacent model outputs and analyzing mean/std/max:
>
> **First-Order Differences Between Adjacent Lead-Time Predictions**
> |**Metrics**|**Vars**|**15–20**|**20–25**|**25–30**|**30–35**|**35–40**|**40–45**|
> |-|-|-|-|-|-|-|-|
> |**mean**|T2m|0.132|0.152|0.187|0.210|0.286|0.316|
> |**mean**|T850|0.199|0.206|0.227|0.239|0.302|0.329|
> |**mean**|Wind10|0.0138|0.0189|0.0196|0.0255|0.0353|0.0531|
> |**mean**|Z500|13.01|13.96|16.47|16.6|18.15|27.9|
> |**std**|T2m|1.973|1.986|2.432|2.596|2.603|2.716|
> |**std**|T850|2.558|2.657|2.921|3.228|3.257|3.527|
> |**std**|Wind10|0.971|1.128|1.18|1.377|1.414|1.497|
> |**std**|Z500|176.2|173.7|203.5|182.3|204.5|226.2|
> |**max**|T2m|11.88|13.01|15.36|16.45|17.57|18.79|
> |**max**|T850|17.43|22.09|27.16|28.22|28.75|28.82|
> |**max**|Wind10|6.852|7.331|8.595|9.913|12.33|12.36|
> |**max**|Z500|1031|1078|1252|1361|1322|1482|
>
> Note: **K-(K+5)** denotes the continuity metric between the K-day and (K+5)-day model predictions.
>
> + **Overall,** the mean, standard deviation, and maximum of the first-order differences increase with lead time across all variables, consistent with the gradual degradation of forecast skill at longer ranges.
> + **At the same time**, the growth of these statistics across adjacent lead-time windows is smooth rather than abrupt, indicating that the fields evolve in a temporally continuous manner instead of exhibiting erratic, frame-to-frame jumps.
>
> ## **Q2: Is this due to the intrinsic difficulty of longer horizons, or does it reflect model collapse?**
>
> Yes, it is due to the intrinsic difficulty of forecasting at longer lead times and reflects model collapse. Specifically, model collapse is itself **one key aspect of the intrinsic difficulty** in longer-horizon forecasting. It manifests as progressive smoothing and loss of detail, arising naturally from multi-step prediction and error accumulation. Beyond this, forecasting at longer horizons also faces other inherent challenges, such as growing uncertainty in atmospheric evolution and the decreasing influence of initial conditions. Our method mitigates the model collapse through noise injection and climatology fusion, thereby** improving performance across extended lead times**.
>
> ## **W5: Minor comments.**
>
> We thank you for your careful reading. We will immediately upload a revised PDF that addresses these minor corrections.

---

> ### Author Response · Authors · 2025-11-25
> **A summary of our revision**
>
> Dear reviewer，
>
> Thanks for your valuable comments. As the next discussion period has opened on November $20^{th}$, we would be grateful if you could allocate some time to review our revision.
>
> We understand that you have a multitude of responsibilities. To facilitate a swift evaluation of our revisions, we have summarized the corresponding changes as follows:
>
> + We conduct an ablation study to compare the impact of injection depth. (Table 4 in the revision).
> + We add a comparison of results on perturbation methods to demonstrate the superiority of our method. (Table 4 in the revision).
> + We add a detailed explanation of method stability and wind forecasting (in Section 4.1).
> + We discuss the consistency analysis and challenges of the model (in Table 11 in Appendix E).
>
> Please let us know if you have any additional concerns or questions. We kindly request that you re-evaluate our paper based on the provided responses and revision. Thank you for your time and consideration!

---

> > ### Comment · Reviewer_FKnz · 2025-11-26
> >
> > Thank you for the detailed response. While I share some of the concerns raised by other reviewers regarding the balance between engineering contributions and academic novelty, your explanations about weaknesses 1 and 4 addressed several of my initial doubts. Overall, I am glad to raise my score to 6 for this important task.
> >
> > I still have one minor question regarding your response to weakness 4. From the reported first-order differences, it is not entirely clear how much of the growth in these statistics reflects intrinsic forecast difficulty at longer horizons versus how much may be attributed to model collapse or temporal discontinuity across lead-specific models. Are there any metrics or diagnostic approaches in S2S forecasting that can help disentangle these two effects? This is not a weakness but rather a question for discussion.

---

> > > ### Author Response · Authors · 2025-11-30
> > >
> > > We sincerely thank the reviewer for raising the insightful questions. We would like to clarify that the model prediction (as shown in our response to weakness 4) is continuous, and the ground-truth physical weather series is fundamentally continuous.
> > >
> > >
> > > ### **How much of the growth in these statistics reflects intrinsic forecast difficulty at longer horizons?**
> > >
> > > We agree that the rate of change is an important factor in forecasting difficulty. First-order differences primarily capture the prediction continuity, which is not able to reflect the forecasting difficulty without comparing to ground-truth data.
> > >
> > > In particular, model collapse, manifested as gradual smoothing and loss of detail, is the key challenge at longer horizons. Intuitively, if the rate of change is abnormal in a certain time step, the model collapse will be accelerated.
> > >
> > >
> > > ### **How much may be attributed to model collapse or temporal discontinuity across lead-specific models?**
> > >
> > > As the above, the model prediction and ground-truth are continuous, the forecasting difficulty is mainly attributed to model collapse.
> > >
> > > ### **Are there any metrics or diagnostic approaches in S2S forecasting that can help disentangle these two effects?**
> > >
> > > Similar to the last question, since the forecasting difficulty is mainly attributed to model collapse, at present, there are no methods to disentangle these two effects. Yet, we agree that the rate of change is related to the model collapse process, which can be investigated by the prediction error and the rate of change of ground-truth. We will add it to the future work, and thank you for providing the valuable insight again.

---

### Official Review · Reviewer_k6Rk · 2025-11-01

**Soundness:** 4
**Presentation:** 4
**Contribution:** 3
**Rating:** 6
**Confidence:** 5

**Summary:**

This paper tackles the challenge of model collapse in Subseasonal-to-Seasonal (S2S) forecasting by introducing a simple yet effective framework, TianQuan-S2S.
The method (1) integrates climatology information into patch embeddings to complement the limited predictive power of initial states, and (2) injects Gaussian noise into Transformer blocks to preserve variability and reduce over-smoothing.
Extensive experiments on the ERA5 dataset show clear improvements over ECMWF-S2S, FuXi-S2S, and ClimaX in both deterministic and ensemble settings.

**Strengths:**

- Well-defined problem motivation: Clearly identifies information loss and model collapse as key S2S challenges.

- Simple but effective design: Climatology integration and noise augmentation yield robust long-lead forecasts.

- Comprehensive experiments: Covers multiple variables, metrics, and spatial resolutions on 40 years of ERA5 data.

- High writing quality and reproducibility: The paper is clearly structured and provides implementation details and code.

**Weaknesses:**

The main limitation of this work lies in its modest technical novelty. The proposed approach largely builds upon existing concepts such as climatology conditioning and stochastic perturbation within Transformer architectures, rather than introducing a fundamentally new modeling paradigm or theoretical insight.

The method can be viewed as a clever and carefully engineered recombination of previously explored ideas, rather than a conceptual breakthrough. While the integration of climatological priors and noise injection is executed elegantly, the individual components are well-known in both numerical and data-driven forecasting literature.

Moreover, the design choices—such as attention-based climatology fusion and Gaussian noise injection at each Transformer layer—feel somewhat heuristic (“tricky”), lacking a deeper theoretical justification for why this specific combination should outperform alternatives like simple concatenation or dropout-based regularization.

Nevertheless, the paper demonstrates clear empirical benefits and provides a well-structured and reproducible evaluation. The proposed model achieves strong, consistent improvements across multiple meteorological variables and lead times, effectively mitigating the long-horizon degradation problem that limits many prior S2S models.

Given that research in subseasonal-to-seasonal forecasting remains relatively sparse, and that the paper delivers a tangible step forward in practical performance and stability, this work deserves publication. It represents a solid and valuable contribution that advances the field through thoughtful system design and comprehensive experimental validation, even if the underlying techniques are not entirely novel.

**Questions:**

The paper introduces a learnable Gaussian noise injection within every Transformer layer, claiming that it helps sustain variability and prevent long-lead model collapse. However, it remains unclear how this mechanism fundamentally differs from existing stochastic regularization techniques such as dropout, layer noise, or even standard input perturbation used in ensemble forecasting.

Specifically, how is the proposed per-layer Gaussian perturbation different from simply perturbing the input fields or initial conditions, as done in conventional ensemble methods like FuXi-S2S or stochastic parameterization in NWP systems? Input perturbations are known to encourage diversity and uncertainty propagation—so what unique advantage does injecting noise throughout the Transformer depth provide?

Moreover, the paper would benefit from a direct experimental comparison between:

- input-only perturbation (ensemble IC perturbations),

- fixed or dropout-style layer noise, and

- the proposed learnable state-dependent Gaussian noise.

Such a comparison could clarify whether the improvement stems from deeper uncertainty modeling or simply from added stochasticity. Additionally, the authors should discuss sensitivity to noise scale and spatial correlation, since uncontrolled noise magnitude might act as a crude regularizer rather than a physically meaningful uncertainty representation.

---

> ### Author Response · Authors · 2025-11-21
> **Author Responses (1/2)**
>
> We sincerely thank the reviewer for recognizing our main contribution and constructive suggestions. We have made every effort to address your comments in the following responses.
> ## **W1&W2: Modest technical novelty & not a conceptual breakthrough.**
> We thank the reviewer for acknowledging the empirical value of our work. As you mentioned, in this work, we identify two crucial technical issues, **insufficient climate modeling and model collapse**, in data-driven S2S forecasting, which have not been investigated before. Although not a conceptual breakthrough, **two simple but effective techniques**, attention-based fusion and learnable noise injection, are introduced into the architecture. This unique, synergistic integration achieves significant improvements in stability and accuracy for S2S forecasts.
>
> In the following, we provide further analysis of these two designs.
> ## **W3&Q: About design choices.**
> We thank the insightful comment regarding the choices of climatology fusion and noise injection. We address them in the following.
> ### **Q1: How does the proposed mechanism differ from existing techniques?**
> Thank you for raising the comparison. Initial condition perturbation (e.g., in FuXi-S2S) only injects noise at the input stage. While this helps generate ensemble spread, its influence diminishes over longer lead times, limiting its ability to prevent model collapse in S2S forecasting[1]. Stochastic parameterization in NWP systems introduces randomness into specific physical schemes[2] (e.g., convection or turbulence), but it is not designed to address over-smoothing in deep learning models.
>
> In contrast, TianQuan-S2S introduces learnable Gaussian noise at every layer of the Transformer. This approach offers two key advantages:
> * The noise is **state-dependent and learnable**, allowing the model to adapt uncertainty injection based on feature representations at different depths.
> * By perturbing every layer, we continuously recalibrate variability throughout the forward pass, effectively **mitigating over-smoothing and maintaining forecast realism** across all lead times.
>
> As suggested by the reviewer, we compare the proposed learnable state-dependent Gaussian noise with (1) Initial-Condition Perturbation; (2) Fixed Noise: remove the learnable functions; (3) Dropout: randomly drops neurons or connections during training. The results are shown as follows.
>
> **RMSE Comparison of Ensemble Mean Results for Different Perturbation Methods**
> |**Variables**|**Perturbation**|**15**|**20**|**25**|**30**|**35**|**40**|**45**|
> |-|-|-|-|-|-|-|-|-|
> |T2m|**Input Perturbations**|2.446|2.506|2.527|2.538|2.556|2.624|2.674|
> |T2m|**Fixed Noise**|2.497|2.556|2.560|2.583|2.601|2.616|2.617|
> |T2m|**Dropout**|2.559| 2.611|2.616|2.617|2.643|2.708|2.764|
> |T2m|**Ours**|**2.424**|**2.457**|**2.459**|**2.502**|**2.471**|**2.532**|**2.601**|
> |Z500|**Input Perturbations**|814|812|809|821|823|808|818|
> |Z500|**Fixed Noise**|825|814|818|827|825|816|823|
> |Z500|**Dropout**|836|830|827|844|843|820|832|
> |Z500|**Ours**|**791**|**779**|**766**|**798**|**805**|**766**|**796**|
>
> The results in the table validate our design, achieving the lowest RMSE for both T2m and Z500 across all forecast horizons (15 to 45 days). Specifically, our method's **continuous variability injection** proves critical for S2S forecasting, reducing the 45-day RMSE over Input Perturbations (T2m by 0.060 K, Z500 by 29 $ m^2/s^2 $).
>
> [1] Stochastic and Perturbed Parameter Representations of Model Uncertainty in Convection Parameterization, 2015.
>
> [2] A Comparison of the ECMWF, MSC, and NCEP Global Ensemble Prediction Systems, 2005.

---

> ### Author Response · Authors · 2025-11-21
> **Author Responses (2/2)**
>
> ### **Q2: What unique advantage does injecting noise throughout the Transformer depth provide?**
> Thanks for the question. While input perturbations are effective in generating initial ensemble diversity, their influence tends to fade as the forecast horizon extends, **limiting their ability to prevent model collapse in long-lead S2S prediction**. Injecting noise throughout the Transformer depth enables continuous uncertainty modeling across the entire forward process. By perturbing every layer, our approach introduces variability at different layers.
>
> To validate this, we conducted an ablation study comparing different noise injection depths. As shown in the table below, applying noise to all Transformer layers consistently achieves the best performance.
>
> **RMSE Comparison of Ensemble Mean Results for Ablation Study of Per-Layer Perturbation**
> |**Variables**|**Injection Layer**|**15**|**20**|**25**|**30**|**35**|**40**|**45**|
> |-|-|-|-|-|-|-|-|-|
> |T2m|**1 Only**|2.599|2.626|2.653|2.714|2.777|2.802|2.841|
> |T2m|**1-2**|2.530|2.562|2.606|2.623|2.693|2.699|2.712|
> |T2m|**1-4**|2.535|2.548|2.593|2.603|2.593|2.652|2.729|
> |T2m|**1-6**|2.511|2.578|2.602|2.567|2.623|2.657|2.707|
> |T2m|**1-7**|2.456|2.496|2.518|2.532|2.554|2.604|2.624|
> |T2m|**All**|**2.424**|**2.457**|**2.459**|**2.502**|**2.471**|**2.532**|**2.601**|
> |Z500|**1 Only**|817|811|806|818|823|807|821|
> |Z500|**1-2**|810|809|798|816|818|792|811|
> |Z500|**1-4**|814|808|802|810|816|785|817|
> |Z500|**1-6**|807|797|772|811|812|779|815|
> |Z500|**1-7**|799|782|776|804|813|778|808|
> |Z500|**All**|**791**|**779**|**766**|**798**|**805**|**766**|**796**|
>
> The ablation study clearly demonstrates that as more layers are perturbed (from 1 Only to All), performance steadily improves and is best sustained at longer lead times. This validates that continuous uncertainty modeling at every block is essential for **preventing model collapse** in S2S predictions.
> ### **Q3: Sensitivity to noise scale.**
> Thanks for the comment. To investigate the sensitivity of our model to these factors, we conducted an additional ablation study. We defined three standard deviations (σ={0.1, 0.5, 1, 1.5, 2}) for the noise applied at each Transformer layer.
>
> **RMSE Comparison of Ensemble Mean Results for Different Noise Scale**
> |**Variables**|**Noise Scale (σ)**|**15**|**20**|**25**|**30**|**35**|**40**|**45**|
> |-|-|-|-|-|-|-|-|-|
> |T2m|**Input Perturbation (σ = 0.1)**|2.509|2.554|2.568|2.569|2.624|2.630|2.718|
> |T2m|**Input Perturbation (σ = 0.5)**|2.477|2.535|2.539|2.543|2.569|2.592|2.679|
> |T2m|**Input Perturbation (σ = 1)**|2.446|2.506|2.527|2.538|2.556|2.624|2.674|
> |T2m|**Input Perturbation (σ = 1.5)**|2.551| 2.580|2.594|2.614|2.664|2.692|2.734|
> |T2m|**Input Perturbation (σ = 2)**|2.611|2.637|2.642|2.652|2.696|2.710|2.774|
> |T2m|**Fixed Noise (σ = 0.1)**|2.507|2.538|2.548|2.553|2.607|2.614|2.696|
> |T2m|**Fixed Noise (σ = 0.5)**|2.505|2.556|2.565|2.583|2.617|2.635|2.707|
> |T2m|**Fixed Noise (σ = 1)**|2.497|2.556|2.560|2.583|2.601|2.616|2.719|
> |T2m|**Fixed Noise (σ = 1.5)**|2.615|2.638|2.648|2.663|2.695|2.717|2.786|
> |T2m|**Fixed Noise (σ = 2)**|2.681|2.708|2.721|2.728|2.762|2.798|2.886|
> |T2m|**Ours (σ = 0.1)**|2.434|2.464|2.474|2.506|2.511|2.561|2.614|
> |T2m|**Ours (σ = 0.5)**|**2.421**|2.463|2.467|**2.496**|2.504|2.558|2.611|
> |T2m|**Ours (σ = 1)**|2.424|**2.457**|**2.459**|2.502|**2.471**|**2.532**|**2.601**|
> |T2m|**Ours (σ = 1.5)**|2.46|2.481|2.495|2.516|2.521|2.564|2.628|
> |T2m|**Ours (σ = 2)**|2.497| 2.506| 2.517|2.533|2.555|2.586|2.639|
>
> Specifically, while all methods show sensitivity to $\sigma$, our learnable noise mechanism is more robust and **reaches its peak accuracy (lowest RMSE)** around $\sigma=1$. Notably, the performance of Input Perturbation and Fixed Noise rapidly drops when the noise scale increases to 1.5, while our method remains stable, demonstrating the effectiveness of learned uncertainty representation.
> ### **Q3: Sensitivity to spatial correlation.**
> Thanks for the comment. We employ the attention mechanism to dynamically capture the spatial variability over climatology. We conduct additional ablation studies with different fusion methods: (1) Concatenation: transforming the concatenation with a linear layer; (2) Learned gate: learning a scalar weight to fuse the input and climatology.
>
> **RMSE Comparison of Ensemble Mean Results for Fusion Methods**
> |Variables|Fusion|15|20|25|30|35|40|45|
> |-|-|-|-|-|-|-|-|-|
> |T2m|Concatenation|2.525|2.556|2.565|2.583|2.627|2.655|2.707|
> |T2m|Gate|2.476|2.528|2.530|2.541|2.563|2.589|2.67|
> |T2m|Attention|**2.424**|**2.457**|**2.459**|**2.502**|**2.471**|**2.532**|**2.601**|
> |Z500|Concatenation|830|814|808|837|845|819|832|
> |Z500|Gate|811|802|797|818|828|802|818|
> |Z500|Attention|**791**|**779**|**766**|**798**|**805**|**766**|**796**|
>
> Our proposed method outperforms baselines at all lead times, confirming that attention is a **more effective and interpretable way to exploit climatological priors**.

---

> ### Author Response · Authors · 2025-11-25
> **A summary of our revision**
>
> Dear reviewer，
>
> Thanks for your valuable comments. As the next discussion period has opened on November $20^{th}$, we would be grateful if you could allocate some time to review our revision.
>
> We understand that you have a multitude of responsibilities. To facilitate a swift evaluation of our revisions, we have summarized the corresponding changes as follows:
>
> + We add a comparison of results on fusion strategies and perturbation methods to demonstrate the ability to model long-term forecast uncertainties. (Table 4 in the revision).
> + We added an ablation study on injection depth, demonstrating the unique advantage of our method. (Table 4 in the revision).
> + We conduct a further study on the impact of noise scale across different perturbation methods (Ours in Figure 6, others in Appendix H, Table 17).
>
> Please let us know if you have any additional concerns or questions. We kindly request your feedback on whether our responses have satisfactorily addressed your concerns. Thank you for your time and consideration!

---

> > ### Comment · Reviewer_k6Rk · 2025-11-25
> >
> > Thank you for your response. All of my concerns regarding the methodology have been resolved. However, I still have some remaining questions about the novelty.
> >
> > As you know, many of the foundational models or core methods in meteorological AI have been developed for medium-range forecasting. Similar to computer vision, where models such as ResNet can be retrained and applied to various domains (e.g., medical imaging or robotics), these medium-range models can also be adapted to different forecasting regimes by simply retraining them in the new domain. From this perspective, GenCast is capable of producing diverse and sharp outputs, and in principle it could also be adapted for an S2S setting.
> >
> > Given this, I am wondering whether your method would still outperform such medium-range models once they are retrained as S2S models. My biggest question is whether your approach is truly specialized and optimized for S2S forecasting in a way that would give it an advantage over adapting existing backbone models to the S2S regime. And if not, I am unsure whether the application-level novelty of achieving SOTA performance in S2S forecasting would be sufficient on its own.
> >
> > To be clear, I am not requesting additional experiments, since I understand that such training would be computationally expensive.
> >
> > If you could help address this remaining concern, I would be willing to raise my score to 8. If not, I would prefer to keep my current score, considering the application-level novelty.

---

> > > ### Author Response · Authors · 2025-11-26
> > >
> > > Thank you for your insightful comments. We are actively working on experiments to address your questions regarding the S2S setting.
> > >
> > > As you noted, it is computationally demanding and requires time. We appreciate your patience and ask that you please bear with us while we complete this essential verification.

---

> > > > ### Comment · Reviewer_k6Rk · 2025-11-26
> > > >
> > > > I agree that additional experiments would be ideal, but in my opinion, sharing practical experience is already meaningful.
> > > >
> > > > From what I know, many agencies routinely retrain models like GraphCast and GenCast for both S2S and medium-range forecasting. If your method clearly outperforms these retrained baselines in such settings, then I don’t see any reason why the paper wouldn’t be strong enough for acceptance.
> > > >
> > > > That’s why I asked—I assumed you had already tried these comparisons when you initially proposed the method.

---

> > > > > ### Author Response · Authors · 2025-11-30
> > > > >
> > > > > Specifically, we can extend the existing medium-range forecasting models to S2S forecasting in the following ways.
> > > > >
> > > > > ### **Tianquan-S2S is specialized for S2S forecasting.**
> > > > >
> > > > > For S2S forecasting, Tianquan-S2S leverages the following designs:
> > > > >
> > > > > * **Incorporate Climatology State**: Extending the medium-range weather forecasting focus on initial condition modeling. However, due to atmospheric chaos, information from the initial state decays rapidly within the S2S period. TianQuan-S2S is specifically designed to overcome this limitation. By incorporating the climatic mean as prior knowledge, it stabilizes predictions within the bounds of historical climate statistics during the initial signal diminution. This approach maintains physical consistency and is essential for enhancing predictive skill throughout the S2S range.
> > > > > * **Efficient uncertainty modeling**: As the forecast time increases, the prediction gradually loses fine-grained scale details, resulting in over-smoothed forecasts. To address this, we efficiently inject learnable, state-dependent Gaussian noise into each layer of the Transformer, demonstrating its empirical effectiveness.
> > > > >
> > > > > ### **Generalizing existing medium-range forecasting to S2S is not straightforward.**
> > > > >
> > > > > Specifically, we can extend the existing medium-range forecasting models to S2S forecasting.
> > > > >
> > > > > * **Autoregression**: A common solution is to iterate the single-step medium-range forecasting models to the S2S timescale. However, its inference complexity is relatively high, and it can lead to significant error accumulation. For verification, we reproduced the 45-day prediction results of GenCast.
> > > > >
> > > > > The experimental setup is as follows:
> > > > >
> > > > > (1) **Data source**: The 2018 ERA5 data from WeatherBench2 was processed as 1-degree data and sampled evenly over 90 days.
> > > > >
> > > > > (2) **Model parameter file**: The trained model parameters of GenCast at 1 degree were downloaded from Google Cloud Bucket.
> > > > >
> > > > > (3) **Settings**: The inference process was implemented on three A800 GPUs. The inference process took 90 steps to obtain the 51 members in the 45-day inference results.
> > > > >
> > > > > (4) **Computation cost**: One inference cost was 5 minutes for one time.
> > > > > |Variables|Model|15|20|25|30|35|40|45|
> > > > > |-|-|-|-|-|-|-|-|-|
> > > > > |T2m|GenCast|2.442|2.532|2.617|2.713|2.759|2.795|2.848|
> > > > > |T2m|**Ours**|**2.424**|**2.457**|**2.459**|**2.502**|**2.471**|**2.532**|**2.601**|
> > > > > |Z500|GenCast|824|831|817|823|829|818|826|
> > > > > |Z500|**Ours**|**791**|**779**|**766**|**798**|**805**|**766**|**796**|
> > > > >
> > > > > As shown in the following table, our method achieved better performance on both the T2m and Z500.
> > > > >
> > > > > * **Joint prediction**: Similar to TianQuan-S2S, another approach is to modify the output head of existing methods to jointly predict the weather on the S2S timescale. Yet, such a diffusion-based generation (analogous to video generation) has long been recognized as a challenging task due to the difficulty of maintaining temporal and physical coherence, and the immense training computational cost and data requirements. In contrast, our method provides an efficient solution to incorporate uncertainty.

---

### Author Response · Authors · 2025-11-21
**General Response to All Reviewers**

Dear Reviewers,

We sincerely thank all the reviewers (`k6Rk`, `FKnz`, `ekwb`) for their valuable feedback. We are glad that the reviewers appreciated the effectiveness and consistency of our proposed framework (`k6Rk`, `FKnz`, `ekwb`), the extensiveness of our experiments (`k6Rk`, `ekwb`), the solidness of experimental results (`k6Rk`, `FKnz`), and the overall quality of our paper's writing (`k6Rk`).

We have made every effort to faithfully address your comments in the responses. As suggested by the reviewers, we add
+ **Additional comparison of results on fusion strategies and perturbation methods**: Included in _Table 4_ in **Section 4.2**, as requested by Reviewers (`k6Rk`, `FKnz`, and `ekwb`) to validate our architectural design choices.
+ **Additional ablation study on injection depth**: Detailed in _Table 4_ in **Section 4.2** to illustrate the impact of noise injection. Raised by Reviewers (`k6Rk`, `FKnz`, and `ekwb`).
+ **Additional ablation study of Impact of the noise scale**: Results are shown in _Figure 6_ in **Section 4.2** and in _Table 17_ in **Appendix H**. Requested by reviewer `k6Rk`.
+ **Additional variables, regional ablation studies, and extreme weather analysis**: Added to **Appendix H** (including _Table 14-16_), specifically addressing the points raised by Reviewer `ekwb`.
+ **Additional explanation of method stability and wind forecasting**: Provided in **Section 4.1**, addressing the concerns raised by Reviewer `FKnz`.
+ **Discussion on Consistency Analysis and Challenge**: Included in _Table 11_ in **Appendix E**, as requested by Reviewer `FKnz`.

We have incorporated the suggested modifications in the revised version, which are highlighted in blue. If you are satisfied with the revisions, we kindly request your approval to consider an improved score.

Thanks for all the reviewers' time again.

Best regards,

Authors

---

### Author Response · Authors · 2025-11-30
**Summary of the Discussion**

Dear Area Chairs, Senior Area Chairs, and Program Chairs,

Our paper initially received ratings of **6 (k6Rk), 4 (FKnz), and 6 (ekwb)**. Throughout the initial discussion period, we systematically replied all reviewers' concerns and substantially improved the manuscript. As a result, the ratings increased to **6 (k6Rk), 6 (FKnz), and 6 (ekwb) (average score 6)** on November $26^{th}$.

And then,
+ Reviewer k6Rk raised an additional question, and said `"If you could help address this remaining concern, I would be willing to raise my score to 8."`
+ Reviewer FKnz wanted to discuss some details which `"is not a weakness but rather a question for discussion."`
+ Reviewer ekwb raised no new questions and said, `"I believe a score of 6 is appropriate for this paper"`.

After the discussion with the reviewers was forcibly terminated, we responded to the above, but it's unclear whether the reviewers' issue has been resolved or if there are any further questions.

## **Reviewer k6Rk (6 → 6)**

Reviewer k6Rk's concerns focused on:

* Clarify whether the improvement stems from deeper uncertainty modeling.
* Discuss sensitivity to noise scale and spatial correlation.
* Whether our approach is truly specialized and optimized for S2S forecasting.

We addressed these by:
+ Providing Supplementary Table 4, which systematically compares different fusion and perturbation methods, to explain our model design.
+ Adding an ablation study on the impact of noise scale. It shows our method maintains greater stability and robustness than other perturbations as the noise magnitude increases.
+ Clarifying why our method can focus on solving S2S forecasting and provides a comparison with GenCast in terms of S2S timescales.

After the first response, reviewer k6Rk stated that `all of my concerns regarding the methodology have been resolved`. We also responded to further questions on whether our approach is truly specialized for S2S forecasting.

## **Reviewer FKnz (4 → 6)**

Reviewer FKnz's concerns focused on:

* Why the injected perturbations are helpful.
* Why the method still helps at shorter leads, and how it prevents model collapse.
* Why wind forecasting is a challenging problem.
* Why the performance decrease does not carry over to TianQuan-S2S.
* No analysis of whether this block boundary introduces temporal discontinuities or not.

We addressed these by:
+ Providing an ablation study on the impact of noise injection depth and perturbation methods in **Supplementary Table 4**.
+ Adding more details on how the response method affects short-term predictions and how the modeling approach prevents model collapse.
+ Discussing the challenges of wind forecasting.
+ Adding new detailed explanations of how our methods hold up well on an S2S scale.
+ Evidencing temporal continuity, the first-order differences show that our forecasts evolve smoothly.

After the first discussion, reviewer FKnz stated that the response `addressed several of my initial doubts`, and **[increased the initial rating 4 to 6](https://openreview.net/forum?id=7Dvmq7MhwU&noteId=9YceayPhxL)** for this important task.

## **Reviewer ekwb (6 → 6)**
Reviewer ekwb's concerns focused on:

* The ablation study of our method on the effects of different variables and regions.
* Need a direct comparison to other uncertainty methods (such as MC Dropout, latent space perturbation).
* How our method performs in challenging conditions.

We addressed these by:
+ Adding an ablation study across different variables and regions to show how our method performs in these settings.
+ Providing **Supplementary Table 4**, which systematically compares the impact of noise depth and different perturbation methods.
+ Adding additional results under these extreme cases (i.e., North America and East Asia heatwaves).

After the rebuttal, Reviewer ekwb **raised no new issues** and maintained the initial positive recommendation (**[score 6](https://openreview.net/forum?id=7Dvmq7MhwU&noteId=zNEavoSKO6)**).

---

### Meta-Review · Area_Chair_ueVq · 2026-01-02

**Summary:**

The paper proposes a method for subseasonal-to-seasonal (S2S) forecasting based on integrating climatology and injecting noise into the transformer blocks to increase the spread. Both techniques are natural/well known, and there are some doubts about the technical novelty. However, the proposed implementation shows strong empirical performance, and there is consensus among the reviewers that the paper presents a valuable addition to the important problem of S2S forecasting. The authors have convincingly addressed some of the initial criticism in their rebuttal (e.g. comparison with alternative noise-injection methods).

**Reviewer Concerns:**

I believe that all concerns about the technical details have been addressed, and the remaining concerns are mainly about the novelty and, to some degree, the (theoretical) motivation of the proposed method. The positive recommendations are mainly based on a thorough evaluation with strong empirical performance on a highly relevant forecasting problem.

**Reviewer Scores:**

I have no reason to doubt what they write in the rebuttal, that the scores would have been raised to 6,6,6

---

### Decision · Program_Chairs · 2026-01-26

Accept (Poster)